# Nucleation of nitric acid hydrates in Polar Stratospheric Clouds by meteoric material

Alexander D. James[1], James S. A. Brooke[1], Thomas P. Mangan[1], Thomas F. Whale[2], John M. C. Plane[1], and Benjamin J. Murray[2]

[1] School of Chemistry, University of Leeds, Leeds LS2 9JT, UK.
[2] School of Earth and Environment, University of Leeds, Leeds LS2 9JT, UK.

*Correspondence regarding PSCs and nucleation experiments to*: Benjamin J. Murray (B.J.Murray@leeds.ac.uk)

*Correspondence regarding meteoric material and WACCM simulations to:* John M. C. Plane (J.M.C.Plane@leeds.ac.uk)

**Abstract.** Heterogeneous nucleation of crystalline nitric acid hydrates in Polar Stratospheric Clouds (PSCs) enhances ozone depletion. However, the identity and mode of action of the particles responsible for nucleation remains unknown. It has been suggested that meteoric material may trigger nucleation of nitric acid trihydrate (NAT; or other nitric acid phases), but this has never been quantitatively demonstrated in the laboratory. Meteoric material is present in two forms in the stratosphere, smoke which results from the ablation and re-condensation of vapours, and fragments which result from the break-up of meteoroids entering the atmosphere. Here we show that analogues of both materials have a capacity to nucleate nitric acid hydrates. In combination with estimates from a global model of the amount of meteoric smoke and fragments in the polar stratosphere we show that meteoric material probably accounts for NAT observations in early season polar stratospheric clouds in the absence of water ice.

## 1 Introduction

Polar Stratospheric Clouds (PSCs) catalyse the activation of Cl and Br species, which in turn catalytically destroy ozone (Solomon, 1999). The phase (liquid droplets or crystals) of PSC aerosol is particularly important: firstly, because rates of activation of Cl and Br species are different depending on the PSC phase; and secondly, because of denitrification where crystals of Nitric Acid Trihydrate (NAT) grow and sediment out of the stratosphere, removing $HNO_3$ and thus preventing formation of the inactive $ClONO_2$ reservoir (Wegner et al., 2012). Nucleation of NAT has been shown to be a major uncertainty in modelling of polar ozone (Brakebusch et al., 2013).

The mechanism and kinetics of nucleation of crystalline PSCs has been a long-standing question (Peter and Grooß, 2012). Formation of NAT on crystalline water ice is known to be a pathway in air masses affected by significant gravity wave activity, which causes rapid cooling. Up to 80 % of denitrification in the Arctic winter of 1999/2000 can be explained by this water ice mechanism (Mann et al., 2005). However, NAT is commonly observed in PSC under conditions where water ice is not stable (and had not been so within tens of hours) which means that there must be an alternative mechanism of NAT nucleation which does not involve ice crystals (Lambert et al., 2016;Voigt et al., 2005).

It is thought that homogeneous nucleation of nitric acid hydrates, either through surface or volume pathways, is not sufficiently rapid to cause observed NAT crystal concentrations in the atmosphere (Knopf et al., 2002). Hence, it has been suggested that heterogeneous nucleation on solid aerosols such as meteoric material might take place. In PSC cases where water ice was not present, an empirical Classical Nucleation Theory (CNT) parameterisation of heterogeneous nucleation on meteoric material gave good agreement with observed clouds (Hoyle et al., 2013). This agreement contrasts with the commonly used approach of assuming a constant nucleation rate in all air masses where NAT is stable, which is able to reproduce some observations but lacks predictive capability since it has no physical basis. Such a predictive capability of nucleation in PSCs and resulting ozone depletion would require characterisation of heterogeneous nucleation kinetics on the appropriate atmospheric material, presumed to be of meteoric origin. While it is often implicitly assumed that NAT nucleates directly, a further complication relevant is that there are several metastable nitric acid hydrates which could potentially nucleate and later transform to the stable NAT phase (Knopf, 2006;Stetzer et al., 2006;Möhler et al., 2006;Grothe et al., 2008;Weiss et al., 2016).

Several previous studies have examined the nucleating ability of analogues for meteoric material. In one case, ground micrometeorite suspended in aqueous nitric acid solutions did not produce observable nucleation (Biermann et al., 1996). The limiting rate of nucleation was not rapid enough to explain PSCs with high number densities, although it was later pointed out that the limiting values were not sufficiently low to rule out nucleation in the atmosphere being important in low NAT number density cases (Hoyle et al., 2013). It has also been noted that meteoric material will at least partially dissolve in acidic droplets, leaving silica materials undissolved (Bogdan et al., 2003;Saunders et al., 2012). It has been found that fumed silica was active as a heterogeneous nucleus of nitric acid hydrates under stratospheric conditions (Bogdan et al., 2003), but no quantification of the atmospheric implications of the measurements was made.

Two forms of meteoric material may be present in the stratosphere, with distinct composition and morphology; Meteoric Smoke Particles (MSPs) and meteoric fragments (MFs). MSPs, which form in the mesosphere from the metallic vapours injected by ablating meteoroids (Plane et al., 2015), are too small to sediment gravitationally and are transported by the residual atmospheric circulation, taking 4-5 years to reach the Earth's surface (Dhomse et al., 2013). As a result of this relatively slow transport MSPs are entrained in sulfate aerosol below 35 km (Neely et al., 2011), causing the dissolution of their metallic components over a period of several weeks (Saunders et al., 2012). It has been shown that the metallic content of meteoric materials is present in solution in mid-latitude stratospheric sulfate aerosol, whilst Si and Al components remain as solid inclusions (Murphy et al., 2014). Amorphous silica is therefore likely to be a useful analogue for MSPs partially dissolved in stratospheric sulfate. Fumed silica is particularly appropriate since its fractal morphology is probably similar to that of MSPs (Saunders and Plane, 2006).

A recent study (Carrillo-Sánchez et al., 2016) using the measured fluxes of Na and Fe in the upper mesosphere, and the deposition rate of cosmic spherules at the South Pole, concluded that $43 \pm 14$ tonnes of cosmic dust enters the atmosphere each day, of which 8 tonnes ablates and goes on to form MSPs. The remaining 35 tonnes of unablated particles, with radii $> 10\,\mu m$, are large enough to sediment rapidly into the troposphere. However, a recent study (Subasinghe et al., 2016) using high

resolution video cameras found that 95 % of meteoroids larger than 1 mm fragmented on atmospheric entry. It was speculated in that study that refractory silicate grains in the meteoroids are held together by a relatively volatile material which evaporates at temperatures significantly below those required to produce thermal ablation of the silicates. This would then result in a significant mass of unablated MFs. If these were submicron size then their slow rate of sedimentation in the lower stratosphere would allow them to influence PSC nucleation. In fact, single particle mass spectrometry measurements of 0.5 wt% Fe in midlatitude sulfate aerosol indicated a meteoric input of 22 to 88 tonnes day$^{-1}$ (Cziczo et al., 2001), suggesting the presence of meteoric material other than MSPs.

Here we investigate whether MSPs and / or MFs could provide a heterogeneous nucleation pathway to NAT in synoptic PSCs where water ice is not present, as shown schematically in Figure 1. Heterogeneous nucleation activities were measured in the laboratory using a drop freezing assay technique (Whale et al., 2015), and then compared to other parameterisations of heterogeneous nucleation in PSCs and atmospheric observations of crystal particle concentrations (Voigt et al., 2005). To carry out these experiments, a range of analogues were chosen for MSPs and MFs. Since MFs are expected to undergo little to no ablation (Taylor et al., 2012), or dissolution of metals in acid droplets (since their sedimentation speed means that they have a short lifetime above the lower stratosphere), they are likely to have compositions similar to interplanetary dust and therefore were represented here by a range of ground meteorite analogues (James et al., 2017). MSPs in the mesosphere can be represented by amorphous materials of olivine composition (James et al., 2017); however, the significant time MSPs spend in acid droplets will cause dissolution of their metallic content (Saunders et al., 2012;Wise et al., 2003). MSPs were therefore represented here by amorphous silica analogues; fumed silica because of its fractal morphology, and fused quartz to test whether the results can be generalised to other forms of amorphous silica. For each of these materials we aimed to determine their ability to nucleate nitric acid hydrates relevant to synoptic PSCs when immersed in binary $HNO_3$-$H_2O$ solutions. This survey across a range of proxies for meteoric material is designed to give an indication of whether MFs or MSPs can nucleate nitric acid hydrates efficiently enough to account for NAT particle observations in the polar stratosphere.

## 2 Methods

Drop freeze assays were performed using a modified version of the Nucleation by Immersed Particles Instrument (NIPI), which we have described previously and has primarily been used to study heterogeneous ice nucleation (Whale et al., 2015). Generally, this technique involves generating aqueous suspensions of particles and then pipetting an array of droplets of known volume onto an appropriate surface on a cold stage. They are then cooled and the freezing point determined optically. The NIPI system has previously been validated by observing well-defined melting points of a variety of organic compounds and ice (Whale et al., 2015). In addition, results from this instrument also compare favourably to a range of other droplet-based techniques for measuring the nucleation efficiency of immersed particles (Hiranuma et al., 2015).

In these experiments, aqueous suspensions of heterogeneous materials were sonicated for one hour to break up aggregated material, shaken by hand (to ensure material was suspended evenly) and immediately used to dilute $HNO_3$ to the desired

concentration. Repeat experiments showed that this method gave reproducible nucleation activities, suggesting that particles were adequately suspended and evenly distributed. A concentration of $39.94 \pm 0.08$ wt% (uncertainties given are 95 % confidence intervals) $HNO_3$ was used, since at stratospheric temperatures this gives saturation ratios ($S_x$, with respect to crystalline nitric acid hydrate phase $x$) relevant to synoptic PSCs. $HNO_3$ concentrations higher than about 45 wt% are not thought to be relevant to the atmosphere in the absence of gravity wave-induced temperature perturbations (Meilinger et al., 1995), whilst lower concentrations led in this experimental setup to nucleation under conditions where water ice is stable, also not relevant to the synoptic clouds we focus on here (Mann et al., 2005). Control experiments were carried out in which samples of $H_2O$ were treated exactly as suspensions, including sonication and mixing with nitric acid. These control experiments were critical for minimising contamination and were routinely performed prior to experiments with the meteoric proxies. By performing control experiments we found that mechanical stirring with magnetic stirrer bars introduced contamination which induced nitric acid hydrate nucleation, hence stirrer bars were not used in these experiments.

Droplets were then pipetted onto a silanised glass slide, surrounded by a spacer (greased Viton o-ring, see Figure 2) and covered with a second glass slide. The restricted volume provided by this sealed cell reduced the possible change in $HNO_3$ concentration due to evaporation or condensation to less than 1 wt% as predicted by a model of equilibrium vapour pressures (Clegg et al., 1998). The cell was situated on an aluminium block with an embedded PT100 thermocouple, on top of the EF600 cooling stage (Whale et al., 2015). Comparison of the EF600 internal temperature probe with this calibrated PT100 show a maximum difference of <1 K at base temperature (183 K). Droplet arrays were made up with the surface held at 291 K to avoid condensation of atmospheric $H_2O$. Samples were then cooled at 5 K $min^{-1}$ to 233 K followed by a 1 K $min^{-1}$ cooling ramp to 183 K and a 3 K $min^{-1}$ warming ramp to 248 K to observe melting. The sample cell was placed inside a larger cell which was flushed with dry $N_2$, zero grade: BOC, during the rapid cooling phase, then sealed to prevent frost formation on the outside of the sample cell. Figure 2 includes a schematic diagram and camera image (room temperature panel).

Phase transitions on warming, which tend to occur without any kinetic limitation, can be a useful way of validating the reported temperatures. On warming, simultaneous visual changes in all the frozen droplets started at $231.2 \pm 0.7$ K and continued until no visible solid was left in the droplets. The temperature at which changes start in the droplet is consistent with either the ice $I_h$ / NAT or NAT / NAM eutectic at 231 K (Martin, 2000) followed by the gradual melting of NAT in equilibrium with a binary $HNO_3$-$H_2O$ solution. Unfortunately, the temperature at which the NAT completely disappeared from the droplets, which would correspond to the NAT-aqueous solution line, was not possible to determine with any confidence due to the optical resolution of the camera, and therefore we do not report it. Nevertheless, the general behaviour is consistent with the established $HNO_3$–$H_2O$ phase diagram (Beyer and Hansen, 2002); this provides further confidence in our reported temperatures.

Samples were selected, and their BET surface areas measured, as analogues for both MSPs (Fused Quartz: $4.85 \pm 0.06$ $m^2$ $g^{-1}$, Fumed Silica: $195 \pm 3$ $m^2$ $g^{-1}$, $MgFeSiO_4$: $208 \pm 2$ $m^2$ $g^{-1}$) and MFs (NWA 2502 (CV3): $5.75 \pm 0.05$ $m^2$ $g^{-1}$, Chergach (H5): $0.44 \pm 0.14$ $m^2$ $g^{-1}$, Allende (CV3): $1.22 \pm 0.15$ $m^2$ $g^{-1}$). Fused quartz powder ($r$ <43 µm) was purchased from Goodfellow Ltd. and used as supplied (without further grinding). Fumed silica (BET surface area 200 $m^2$ $g^{-1}$) was purchased from Sigma and

used as supplied. MgFeSiO4 was synthesised from chemical precursors using a sol-gel method we have described previously (James et al., 2017), and ground on an agate pestle. Sample analogues are summarised in Table 1.

These samples were chosen in order to cover a range of stratospherically relevant particle properties: chemical and crystalline characteristics of MSPs (MgFeSiO4), the fractal agglomerate morphology of MSPs (fumed silica), and the chemical composition of MSPs after partial dissolution in acid droplets (fused quartz and fumed silica). SiO2 materials were characterised in terms of their crystalline, morphological and elemental composition. Very limited information is available from the manufacturers of these materials. Powder X-ray diffraction was used to confirm that the two SiO2 materials contain no measurable crystalline component, as shown in the Supplementary Information (SI; see figure S1). Scanning Electron Microscopy with Energy Dispersive Electron spectroscopy (SEM-EDX, see figures S2 and S3) showed that the fused quartz consists of particles ranging from several hundred nm to several µm diameter, composed of Si and O only (detection limit of 0.1 %). Fumed silica was found to have a fractal morphology, made up primary particles of just ~6 nm, similar to MSPs and previous microscopy images of fumed silica (Bogdan et al., 2003). MSP analogue suspensions were made up with concentrations of $2.512 \pm 0.019$ wt% of each analogue.

Meteorite samples were purchased from meteorite-market.com. Detailed characterisation of the Chergach and Allende meteorites can be found in our previous work (James et al., 2017). 8 g of the North West Africa (NWA) 2502 meteorite was ground in an agate ball mill for 1 hour. Chergach (1 g) and Allende (4 g) samples were ground with an agate pestle and mortar. MF analogue suspensions were made up with concentrations of $0.043 \pm 0.002$ wt% of each analogue.

Fumed silica is manufactured by flame pyrolysis of chemical precursors, typically SiCl4, and was selected as an analogue for MSPs with likely similar fractal morphology. To investigate the possibility that other amorphous silica materials might nucleate with different efficiency, fused quartz, which is manufactured by melting and flash freezing of quartz or silica, was also used. MgFeSiO4 particles synthesised in our laboratory (James et al., 2017) and ground on a pestle and mortar were used as an analogue for MSPs without significant alteration by acid droplets.

The three meteorite samples were chosen as analogues for MFs since they contained fine-grained particles of relevant phases (Taylor et al., 2012;James et al., 2017). The Allende CV3 meteorite has been used widely in the literature as an analogue for interplanetary dust and ablating meteors (Gómez-Martín et al., 2017;Clarke et al., 1971); it is distinct in that it contains significant calcium / aluminium inclusions. NWA is also a carbonaceous CV3 type meteorite and is known to contain significant magnetite (Russell et al., 2005), identified as a coating of micrometeorites (Biermann et al., 1996). Chergach is an H5 ordinary chondrite and was included here to investigate the possibility that the activity of ground meteorites was due to the organic component; it is also known to contain significant metallic Fe and Ni (Weisberg et al., 2008).

**3 Results**

Nucleation of crystalline phases was observed at temperatures relevant to PSCs in droplets of $39.94 \pm 0.08$ wt% HNO3 in H2O. Figure 2 shows the evolution of an array of droplets during an experiment. Nucleation events were observed between 210 and

183 K during cooling by a clear 'darkening' as crystals grew within the droplets. The crystallisation of the droplet occurred in less than one second (the image acquisition rate) to over several seconds and nucleation was taken as the first sign of crystal growth.

A heterogeneous effect on nucleation was observed when analogues for MSPs or MFs were suspended in the solution, as shown in Figure 3. The fused quartz and NWA meteoritic samples were found to nucleate at significantly warmer temperatures than other samples, with some nucleation observed above the melting point of nitric acid dihydrate (NAD), $T_{NAD}$. Nucleation events observed above $T_{NAD}$ can only be achieved by direct nucleation of NAT or of some hitherto unknown nitric acid hydrate since no other solids are thermodynamically stable.

In order to test whether dissolved metal salts, which would be present in atmospheric droplets, could alter the nucleation activity of suspended particles, salts of $Fe^{3+}$ and $Mg^{2+}$ were added to suspensions of fumed silica. No effect on the nucleating temperature was observed. This is in contrast to an earlier study (Wise et al., 2003), which found that freezing of aqueous sulfuric acid solutions increased by up to 10 K when metal salts were added. The authors in that case speculated that soluble $Fe^{3+}$ or a combination of that with other metal ions affected the nucleation process. The lack of a similar effect here could be a result of working in a different acid solution, nucleating a different phase or the differing volume of samples. That study also differs from this in that our experiments include particles which control the nucleation and may have active sites which are not susceptible to the chemical effects of the dissolved metals.

The fraction of droplets which nucleate on cooling to a temperature $T$, $f(T)$, depends on the material which is present and the available surface area, $s$, of that material in each droplet. Hence, fraction frozen is an experiment specific quantity and cannot be directly related to freezing in the atmosphere. To facilitate application to PSCs and also assess the differing activity of each material, the cumulative number of sites per surface area active under given conditions of $S_x$, $n_s$, was calculated by $n_s = -\frac{ln(1-f(T))}{s}$ (Murray et al., 2012). In this approach it is assumed that the number of nucleation events is primarily controlled by $S_x$ (which in our experimental data at constant concentration is in turn determined by temperature) of the system and that the time dependence of nucleation is of secondary importance. This approximation works well when the active sites across a surface are diverse in their ability to nucleate (Herbert et al., 2014). Testing the relative importance of time dependence of nucleation by the active materials identified here, using the methodology developed by Herbert et al. (2014), would be a useful topic of future work. An alternative, but more complex, approach is to define the spectrum of nucleation sites within the context of CNT where each site has a characteristic contact angle and the freezing probability is summed over the distribution of sites (Hoyle et al., 2013;Niedermeier et al., 2011;Broadley et al., 2012). Hoyle et al. (2013) used such a model to show that when $S_{NAT}$ in the atmosphere did not change, the number of NAT crystals only increased marginally. Hence, using $n_s$ to extrapolate experimental data to atmospheric contexts will lead to a slight underestimation of the concentration of NAT particles produced in a cloud.

Figure 4 shows that the materials investigated here have a wide range of activity ($n_s$) over atmospherically relevant conditions of $S_{NAT}$. The three ground meteoritic samples (which were selected as they contain a range of different minor component phases, see Sect. 2 and Table 1) have remarkably similar activity when plotted in $n_s$ despite nucleating at significantly different temperatures. This suggests that the major olivine / pyroxene phase may dominate the heterogeneous activity, rather than the

carbonaceous, metallic or any other minor mineral component that varies strongly between the three samples. The apparent agreement between the active MSP analogue fused quartz and the meteoritic samples may be coincidental, since the silicon content of meteorites is dominated by crystalline olivine and pyroxene, whilst silica minerals or amorphous silica phases are negligible (Jessberger et al., 2001). The fumed silica and MgFeSiO$_4$ materials show remarkably lower activity compared to the other samples. The reasons for this differing activity are unclear, but we discuss three possible explanations. First, relatively

small changes in surface properties of materials may have significant impacts on their nucleating activity. This was recently observed for alkali feldspars where alkali feldspars with very similar crystal structures and compositions have very different abilities to nucleate ice (Harrison et al., 2016;Whale et al., 2017). It is thought that surface features associated with strain in the crystal structure are associated with sites on which nucleation occurs (Whale et al., 2017). Second, the grinding process might introduce features to the surfaces which cause nucleation (Hiranuma et al., 2014), and indeed several of the more active

samples were ground. However, the MgFeSiO$_4$ sample was ground on a pestle and mortar but remains relatively inactive, hence the grinding process alone cannot account for the observed differences in activity. Third, the significantly smaller activity of the fumed silica and MgFeSiO$_4$ may be a result of their morphology, specifically the particle size. Both have specific surface areas around 200 m$^2$ g$^{-1}$, corresponding to an equivalent spherical radius of ~6 nm (Bogdan et al., 2003;James et al., 2017). Such small particles are of a similar order to the size of critical clusters and according to classical nucleation theory are

therefore thought to be relatively poor at causing nucleation (Pruppacher and Klett, 1978). In contrast fused quartz has a specific surface area of 4.85 m$^2$ g$^{-1}$, corresponding to spherical particles of 234 nm radius, likely much bigger than the critical cluster. This might suggest that MSPs, which have nanometer scaled primary grains, are less effective at nucleating nitric acid hydrates than MFs which will tend to be significantly larger. Separating the effect of size and inherent nucleating ability should be the focus of a dedicated future study.

In order to assess the atmospheric implications of these observations, the assumption has been made that the nucleation events observed in this study were direct nucleation of NAT. While observations indicate that NAT is the phase which exists in PSC (Höpfner et al., 2006), it is possible that other metastable nitric acid hydrate phases (Nitric Acid Dihydrate, α- or β-NAD) may form initially, then transform to the stable NAT phase (Grothe et al., 2008;Weiss et al., 2016). We note that the 820 cm$^{-1}$ feature used by Höpfner et al. (2006) to identify atmospheric NAT is present for both the α- and β- polymorphs (Iannarelli and

Rossi, 2015). Since the equivalent 816 cm$^{-1}$ feature for β-NAD has not to our knowledge been compared to the atmospheric spectra there is still uncertainty regarding the relevant atmospheric phases. In fact, NAD nucleation has been observed under atmospheric conditions for homogeneous nucleation (Stetzer et al., 2006). However, in our experiments some nucleation events were observed under conditions where NAD is not thermodynamically stable ($S_{NAD} < 1$), and since there is no significant

discontinuity in the trend in $n_s$ at $S_{NAD} = 1$, the assumption of direct nucleation of NAT seems reasonable. Since it is possible that a different nitric acid hydrate phase formed in these experiments we have examined the sensitivity of our atmospheric conclusions to the assumption of NAT as the primary nucleating phase. Some metastable NAD may form when $S_{NAD} > 1$ (or some other metastable nitric acid phase may form); however, the consistent melting onset of droplets in agreement with the

NAT / ice or NAT / NAM eutectic suggests that if any NAD did form it converted to NAT (note that the melting point is not taken to be supporting evidence of which phase nucleated initially). We did not attempt to identify directly the phase of the acid hydrate in the frozen droplets, since the polymorph resulting from crystallisation may not be the same phase which initially nucleated. In fact, if a metastable phase nucleates, it often converts to a more stable phase during the crystallisation process (Murray and Bertram, 2008). The parameterisations of $n_s$ as a function of $S_{NAT}$ shown in Figure 4 were therefore used to

investigate the activity of meteoric material in heterogeneously nucleating PSC formation in the atmosphere.

## 4 Implications for stratospheric clouds

In order to assess the heterogeneous nucleation of PSCs by meteoric material, it is necessary to estimate the availability of MFs and MSPs, and the conditions (temperature and concentration of droplets) at the time of nucleation.

### 4.1 Availability of meteoric material in the stratosphere

Simulations of MSP growth and sedimentation have been carried out in a number of previous studies (Bardeen et al., 2008;Frankland et al., 2015;Neely et al., 2011). For this study we used atmospheric modelling data from (Brooke et al., 2017). These were Whole Atmosphere Community Climate Model (WACCM, e.g., Marsh et al. (2013)) runs with specified dynamics using the Modern-Era Retrospective Analysis for Research and Applications (MERRA) reanalysis (Rienecker et al., 2011), an MSP input of 7.9 t d$^{-1}$ (Carrillo-Sánchez et al., 2016), and included an interaction between MSP and sulfate aerosol through

condensation and heterogeneous nucleation (see supplementary material for details). Aerosol microphysics calculations were performed using the Community Aerosol and Radiation Model for Atmospheres (CARMA) model (Toon et al., 1988;Toon et al., 1979;Turco et al., 1979). The zonal mean MSP density at 67° N and 70 hPa during February, averaged over 2011-2014, is $(1.5 \pm 0.5) \times 10^{-15}$ g cm$^{-3}$.

If these MSPs were spread evenly among liquid droplets with a concentration of 20 cm$^{-3}$ (number per unit volume of

atmosphere), and agglomerate or partially dissolve / precipitate to form one sphere in each droplet, they would have a radius of 20 nm and a total surface area of $0.17 \pm 0.04$ µm$^2$ cm$^{-3}$ (surface area of MSP per unit volume of atmosphere). This is a factor of 30 times smaller than assumed in a previous study (Biermann et al., 1996), while in a more recent modelling study (Hoyle et al., 2013) the assumption of 7.5 cm$^{-3}$ foreign spheres of 20 nm radius results in a surface area around three times smaller than that estimated here. Observations have revealed around 8 cm$^{-3}$ refectory particles in the polar vortex (Weigel et al., 2014).

Since the available surface area of MFs is much more poorly constrained, the surface area of MSPs has been scaled according to the mass ratio of ablated to unablated material (giving a surface area of 0.76 µm$^2$ cm$^{-3}$, a factor of 4.5 larger than MSPs) (Carrillo-Sánchez et al., 2016). If fragmentation reduced the size of meteoroids to that of MSPs then they would be subject to

atmospheric circulation and focussing in the polar vortex, and would therefore have significant lifetimes in acid droplets and partially dissolve. However, estimates of meteoric input fluxes from the measurements in the stratosphere suggest that the input flux may be higher than that estimated here by a factor of two (Cziczo et al., 2001). The assumption that meteoroids fragment to MSP sizes will produce a larger bias than this uncertainty in mass, so this estimate of surface area is likely an upper limit but is considered physically reasonable for the purposes of this study.

### 4.2 Atmospheric conditions and impact of heterogeneous nucleation

The nucleation of NAT has been shown here to be extremely sensitive to $S_{NAT}$, which depends on the temperature and chemical composition of liquid droplets (Clegg et al., 1998). Several case studies, using time independent temperature or an observed atmospheric temperature profile, are now used to investigate the relevance of the measured nucleation activity in the atmosphere. Atmospheric droplet concentrations and $S_x$ are calculated from the Aerosol Inorganic Model (Clegg et al., 1998), and in combination with our experimentally derived $n_s(S_{NAT})$ the resulting concentration of crystals in a given atmospheric volume, $N_{NAT}$, are derived. Availability of meteoric materials was taken to be as described above and the initial number concentration of liquid droplets was set to 20 cm$^{-3}$. Since virtually all H$_2$SO$_4$ at PSC altitudes condenses onto droplets at temperatures above 200 K, and H$_2$O does not affect $S_{NAX}$ until below 194 K, equilibrium droplet concentrations and $S_{NAX}$ when nucleation occurs are most sensitive to the available mixing ratio of gas-phase HNO$_3$.

Figure 5 illustrates the predicted $N_{NAT}$ as a function of atmospheric temperature for typical stratospheric mixing ratios of H$_2$SO$_4$, HNO$_3$ and H$_2$O. The plot shows that the MF analogues and the more active MSP analogue have sufficiently large $n_s$ to produce in excess of the atmospheric abundances of NAT crystals at a supercooling of several degrees below the NAT melting point ($T_{NAT} \sim 198$ K) and at temperatures well above the frost point (i.e. where water ice cannot form). This suggests that both MFs and MSPs have the potential to nucleate NAT in the stratosphere. However, the less active MSP analogues cannot explain the observed number concentrations: determining the applicability of each of these amorphous silica materials as MSP analogues and why some nucleate extremely efficiently while others do not, are topics for future work.

Figure 5 also shows that varying the atmospheric concentration of HNO$_3$ within observed limits makes a significant difference to the nucleation activities predicted by the parameterisations from the present study. $S_x$ in a droplet in equilibrium with the gas phase increases approximately linearly when the available HNO$_3$ is varied from 10-15 ppbv (assuming 5 ppmv H$_2$O and 0.1 ppbv H$_2$SO$_4$). However, this produces a logarithmic increase in $n_s$, which significantly increases $N_{NAT}$.

While Figure 5 shows that meteoric materials have the potential to trigger nucleation of NAT under stratospheric conditions, atmospheric trajectories of air parcels tend to be more complex. Figure 6 illustrates the NAT concentration expected for the temperature history taken from an observed cloud (Voigt et al., 2005), where only a modest $S_{NAT}$ (<20) was reached and water ice could not have formed. The equilibrium droplet concentration and resulting $S_x$ were calculated as a function of time for the back trajectory temperatures from the atmospheric measurement (Voigt et al., 2005), assuming a range between 10 and 15 ppbv HNO$_3$. The temperature profile and resulting $S_{NAX}$ are shown in Figure 6 (a). The resulting $N_{NAT}$, Figure 6 (b),

were then calculated and compared to the atmospheric observations corresponding to the temperature profile. The $n_s$ ($S_{NAT}$) parameterisations measured here for fused quartz and meteorites are able to reproduce the observed density of NAT crystals.

In Figure 6b, we compare to the constant volume nucleation NAT production rate used by Voigt et al. (2005) and also the CNT based model of Hoyle et al. (2013). These literature models have been tuned to reproduce observed NAT concentrations and so generally produce similar $N_{NAT}$, but the production of NAT earlier in the cloud's evolution vary markedly. The constant volume nucleation rate (here $8 \times 10^{-6}$ cm$^{-3}$ hour$^{-1}$, taken from Voigt et al. (2005)) produces a steep increase in NAT crystal number at relatively low $S_{NAT}$. In contrast, the NAT production based on the $n_s$ ($S_{NAT}$) parameterisations is much more gradual (it should be noted that the $n_s$ ($S_{NAT}$) parameterisation is extrapolated to $S_{NAT} = 1$, whereas in reality $n_s$ will tend to zero on approaching $S_{NAT} = 1$). The delay in production of NAT to higher $S_{NAT}$ is consistent with modelling which shows that a temperature bias of 1.5 K on all aerosol processes was found to better reproduce observed ozone when using a constant volume nucleation parameterisation (Brakebusch et al., 2013), although this is likely to arise in part because of biases in the model temperature fields (Solomon et al., 2015). The $n_s$ parameterisations measured here tend to predict nucleation of NAT earlier and with a more gradual increase than the CNT approach used by Hoyle et al. (2013). Each of these differences could have significant implications for predicted Cl and Br activation, and ultimately ozone depletion.

To test the sensitivity of the system to the assumption that NAT was the primary nucleating phase, parameterisations of $n_s$ as a function of $S_{NAD}$ were produced for the meteorite samples and the fumed silica. Note that a parameterisation was not produced for the fused quartz since with this material heterogeneous nucleation was always observed under conditions too warm (by up to 10 K) for NAD to be thermodynamically stable. These parameterisations were then implemented in the same atmospheric scenarios (temperatures and concentrations). The result was that the onset of nucleation was predicted 1.2-1.7 K colder, but that by the point of measurement around 250 times higher $N_{NAX}$ was predicted (data not shown). This means that the conclusions remain the same for both the meteorites (which are sufficiently active to explain observed cloud) and fumed silica (which is not), suggesting that the atmospheric conclusions of this study are reasonably insensitive to the choice of primary nucleating phase.

One major uncertainty, which should be a topic of future work, is the impact of $H_2SO_4$ on the nucleating ability of these materials. MSPs in particular will be included into $H_2SO_4$ droplets significantly above the altitudes where PSC form and are thought to be processed chemically in those droplets (Neely et al., 2011;Saunders et al., 2012). Whilst we have used analogues for MSPs after processing, a more complete study would work in ternary solution droplets of atmospherically relevant concentrations. Since covering a significant portion of the phase diagram would be extremely time consuming, and since the methods used here would need to be significantly modified to investigate chemical processing at relevant temperatures, no work on ternary solutions is included here. Further work quantifying the time dependence of nucleation and the availability of MFs in the stratosphere would also allow the production of a parameterisation which could be used to investigate heterogeneous nucleation of PSCs in global models. Neverthless, we demonstrate here - for the first time - that meteoric materials have the potential to nucleate NAT particles under stratospheric conditions.

## 5 Conclusions

Heterogeneous nucleation by analogues for Meteoric Smoke Particles (MSPs) and Meteoric Fragments (MFs) in binary $HNO_3$ / $H_2O$ solutions has been measured in the laboratory. Both MSPs and MFs immersed in nitric acid solution droplets were found to nucleate crystalline nitric acid. Given nucleation occurred under conditions where the metastable nitric acid
dihydrate (NAD) is unstable, we suggest that the nitric acid trihydrate (NAT) initially nucleated, although we cannot rule out the nucleation of other metastable phases. Parameterisations of the resulting activity were used to show that heterogeneous nucleation on meteoric material is a potential pathway to forming observed number densities of NAT crystals in Polar Stratospheric Clouds (PSCs) in the absence of water ice.

Good agreement between three significantly different meteorites suggests that the nucleating ability is controlled by the bulk
olivine / pyroxene phases common to each sample. Striking differences were observed between the amorphous silica analogues, possibly due to morphological differences. We note that all materials with very small primary grain sizes (with very large specific surface areas) were relatively ineffective at nucleating NAT. This is consistent with classical nucleation theory which predicts that as a particle approaches the size of a critical cluster its nucleating ability decreases sharply. This might indicate that MSPs, which have small primary particle sizes, may not be very effective at nucleating NAT in the atmosphere,
whereas MFs are likely composed of larger grains and might therefore nucleate NAT more effectively. However, further work is required to separate the effect of small particle size from inherent nucleating ability in order to quantify heterogeneous NAT production in the polar stratosphere.

The parameterisations developed here predict nucleation at cooler temperatures than parameterisations based on a constant nucleation rate per volume, but at warmer temperatures than a recent parameterisation based on classical nucleation theory.
All parameterisations were found to be sensitive to assumptions of the available $HNO_3$. The availability of MFs is also a major uncertainty and future work on PSCs would clearly benefit from a quantified input of meteoric fragments.

It has been shown here that meteoric material, both MSPs and MFs, can trigger nucleation in PSCs. This could have significant implications for future modelling of PSC formation, Cl and Br activation (through differing rates of activation and denitrification) and ultimately ozone destruction.

**Data and sample availability**

Raw experimental data, fit to Voigt et al. (2005) temperature profiles and data relating to subsequent plots in Sect. 4 are available at https://doi.org/10.5518/312. The available surface areas assumed for MSPs and MFs were taken from a WACCM run which is described in the supplementary text, see Brooke et al. (2017) for further description and data availability. Samples of materials used may be obtained (subject to availability) on request from JMCP, or see James et al. (2017) for further details
of the origins and preparation of suitable alternatives.

**Supplement Links**

 [Link to supplementary text to be inserted by copernicus]

**Author contributions:**

ADJ performed all experiments and data analysis and led authorship of this manuscript.

JSAB supplied modelling data which allowed interpretation of the atmospheric implications of the laboratory measurements.

TPM assisted with data analysis and interpretation.

TFW advised on selecting silica material analogues and assisted in developing the experimental method.

JMCP assisted with estimates of atmospheric concentrations of meteoric material and supervised the project.

BJM supervised experiments and contributed significantly to interpreting data both in the laboratory and atmospheric contexts.

**Competing interests**: The authors declare no competing financial interests

**Acknowledgements**

This study was funded by the European Research Council (project number 291332 – CODITA; 240449 – ICE; 632272 – IceControl; 648661 – MarineIce; 713664 – CryoProtect). We would also like to thank Prof. Martyn Chipperfield and Prof. Margaret Campbell-Brown for insightful discussions and Dr. Wuhu Feng for his long term involvement in developing WACCM and useful comments on the discussion paper.

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

**Tables**

Table 1- Description of analogue samples.

| Analogue | Represents | Specific surface area / $cm^2$ $g^{-1}$ | In house processing | Comments |
|---|---|---|---|---|
| Fused Quartz | MSPs | $4.85 \pm 0.06$ | None | Produced by melting and flash freezing quartz |
| Fumed Silica | MSPs | $195 \pm 3$ | None | Fractal agglomerates. Produced by flame pyrolysis |
| $MgFeSiO_4$ | MSPs | $208 \pm 2$ | Hand Ground | "Folded sheets" morphology (James et al., 2017) |
| NWA 2502 | MFs | $5.75 \pm 0.05$ | Ball Milled | CV3 - Contains magnetite |
| Chergach | MFs | $0.44 \pm 0.14$ | Hand Ground | H5 - Low carbonaceous content |
| Allende | MFs | $1.22 \pm 0.15$ | Hand Ground | CV3 - Contains calcium / aluminium inclusions |

**Figures**

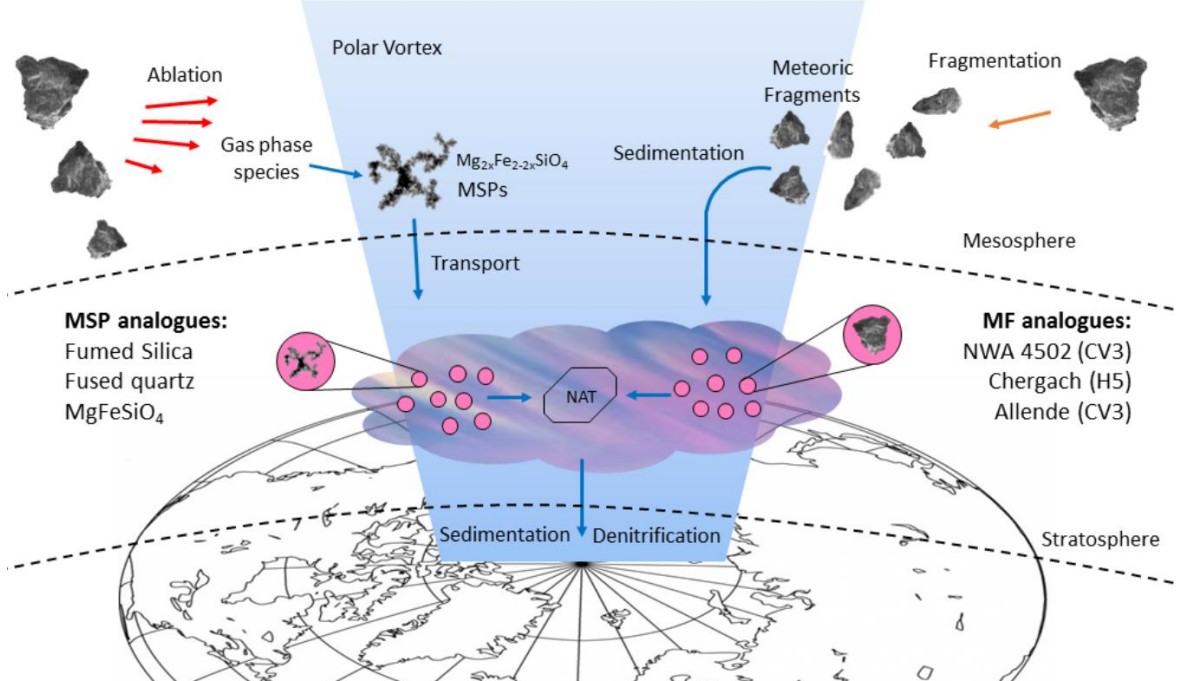

**Figure 1: The pathways of two kinds of meteoric material through the atmosphere. Processes are shown which lead to formation of Meteoric Smoke Particles (MSPs) and Meteoric Fragments (MFs), either of which could heterogeneously nucleate Nitric Acid Trihydrate (NAT) in polar stratospheric clouds.**

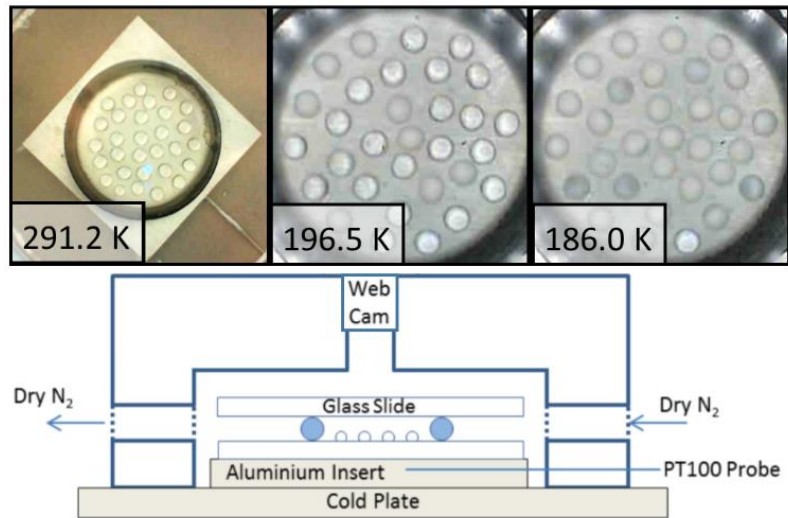

**Figure 2: Droplet array on cooling, demonstrating that crystallisation and therefore nucleation temperatures can be distinguished. Panel at room temperature shows the supporting cold plate, aluminium insert with PT100 temperature probe and slide / spacer / slide cell configuration. Panels below room temperature show nucleation events for a 39.98 wt% $HNO_3$ solution with 0.045 wt% of ground Chergach meteorite particles in suspension (in one droplet shown in the lower right, no nucleation was observed even at 186 K). A side on schematic diagram of the apparatus is also shown (see Sect. 2).**

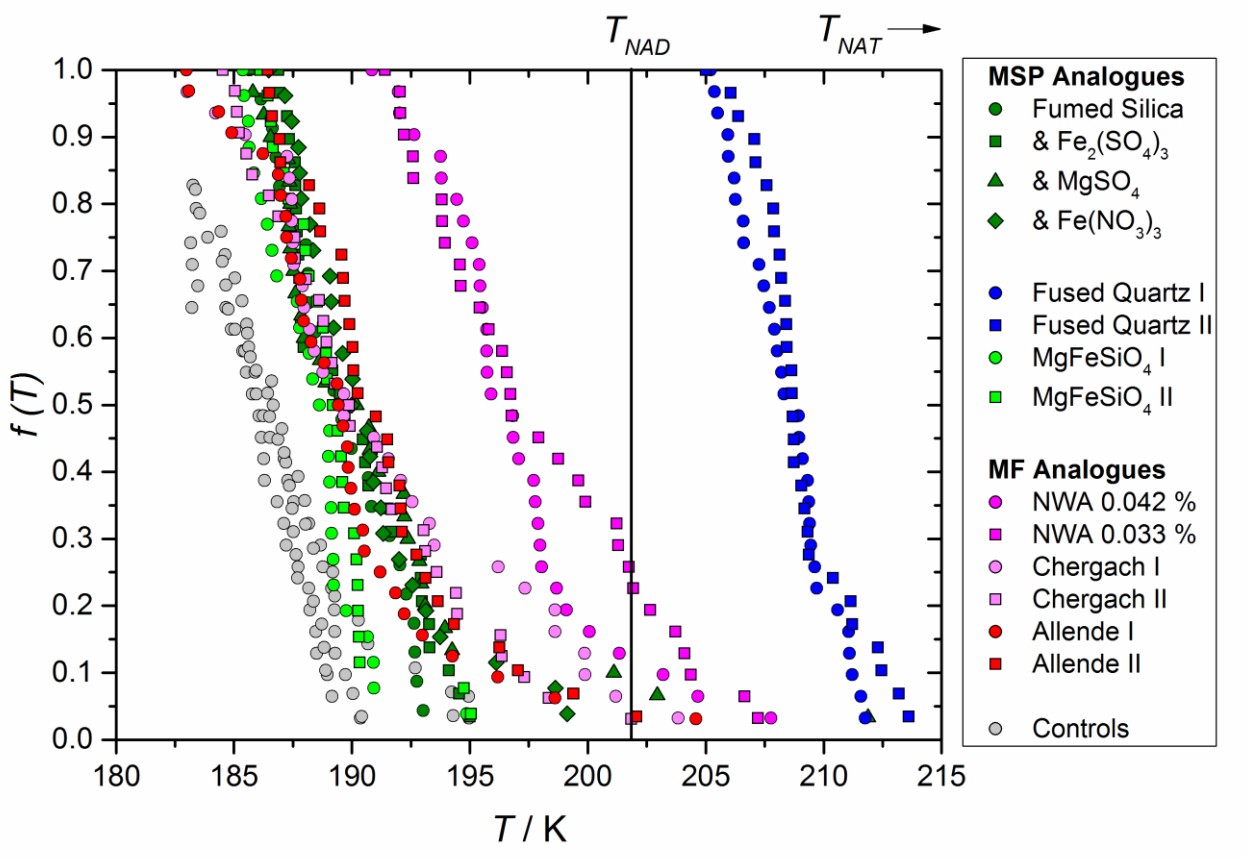

**Figure 3: Fraction of droplets crystallised at different temperatures showing a heterogeneous effect for a variety of analogues for MSPs and MFs. Experiments were carried out using particulate solids suspended in 39.94 ± 0.08 wt% HNO₃ in H₂O. MSP analogues are shown in blue and green colours and were suspended to concentrations of 2.512 ± 0.019 wt%, MF analogues are shown in pink and red colours and were suspended in concentrations of 0.043 ± 0.002 wt% except where stated in the legend for the NWA meteorite. Repeat experiments, distinguished by roman numerals, are shown as different symbol shapes. Experiments with fumed silica labelled with metal salts contained 0.635 ± 0.005 wt% of the relevant metal ion as well as fumed silica. The instrument background (when no heterogeneous material was added to the nitric acid solution) measured in four repeat experiments is shown as grey points. A vertical line is used to show the NAD melting point ($T_{NAD}$), whereas the NAT melting point ($T_{NAT}$) was at 244 K.**

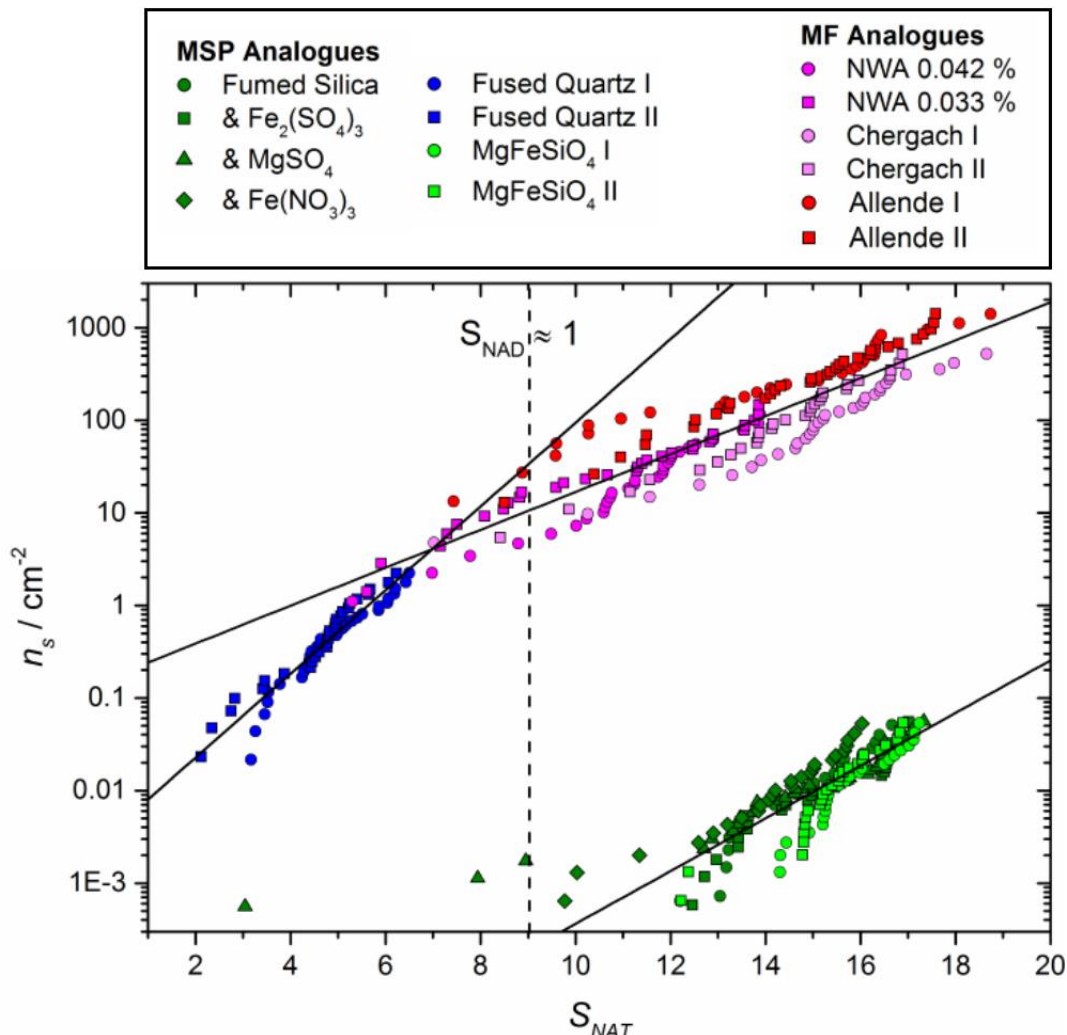

**Figure 4: Nucleation activity of MSP and MF analogues investigated here expressed as active sites per unit surface area ($n_s$) as a function of $S_{NAT}$. The vertical dashed line shows the approximate $S_{NAT}$ at which NAD becomes stable. Samples were selected and their BET surface areas measured as analogues for both MSPs (Fused Quartz: $4.85 \pm 0.06$ m$^2$ g$^{-1}$, Fumed Silica: $195 \pm 3$ m$^2$ g$^{-1}$, MgFeSiO$_4$: $208 \pm 2$ m$^2$ g$^{-1}$) and MFs (NWA 2502 (CV3): $5.75 \pm 0.05$ m$^2$ g$^{-1}$, Chergach (H5): $0.44 \pm 0.14$ m$^2$ g$^{-1}$, Allende (CV3): $1.22 \pm 0.15$ m$^2$ g$^{-1}$). Log-linear fits have been added as a guide to the eye and to facilitate comparison with atmospheric conditions.**

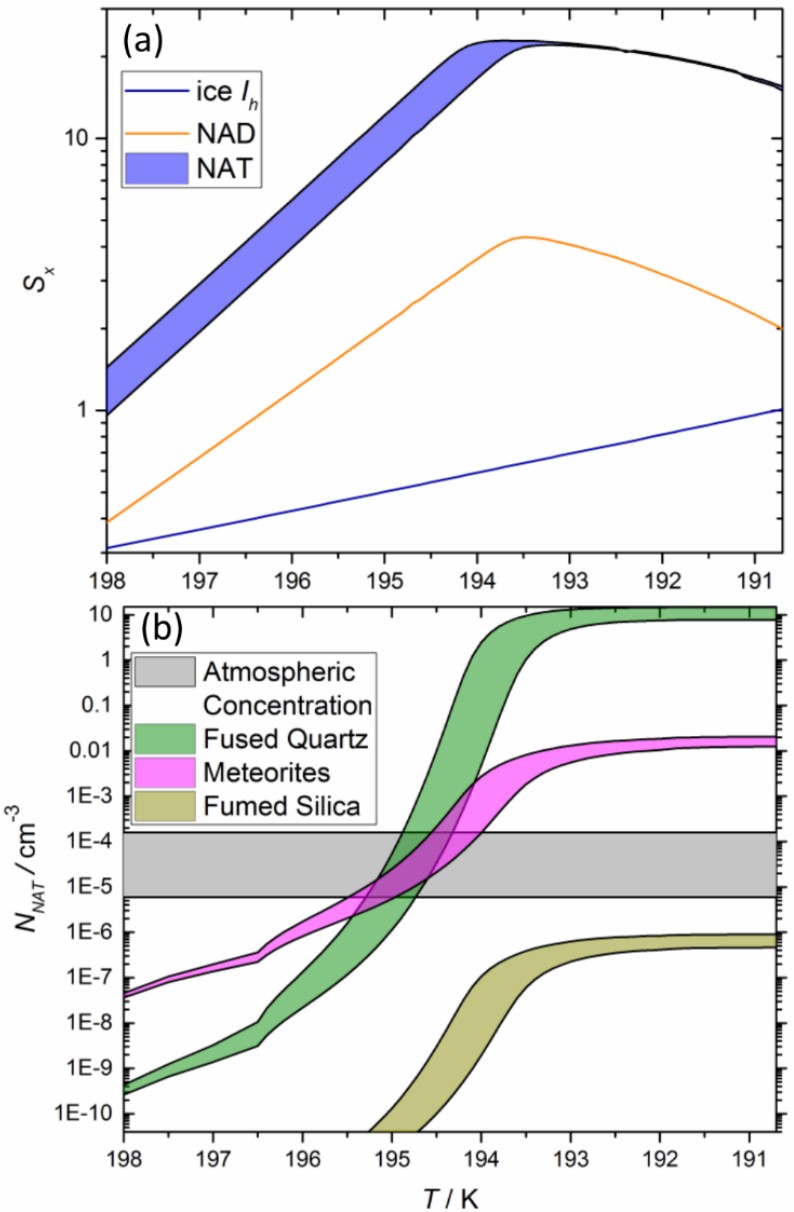

**Figure 5: Plot demonstrating that the parameterised nucleating activities of meteoric materials are able to reproduce typical cloud NAT particle concentrations.** Saturations with respect to relevant crystalline phases, $S_x$, are shown in panel (a) and number concentration is shown in panel (b). Equilibrium $S_x$ were calculated for air masses at 70 hPa (approximately 18 km) containing 5 ppmv $H_2O$, 0.1 ppbv $H_2SO_4$ and either 10 (minimum of ranges) or 15 ppbv (maximum of ranges) $HNO_3$ using the Aerosol Inorganic Model (Clegg et al., 1998), and number concentrations were calculated using the $n_s$ parameterisations shown in Figure 4, assuming 20 liquid droplets $cm^{-3}$ air and the availability of meteoric materials described in Sect. 4 of the text.

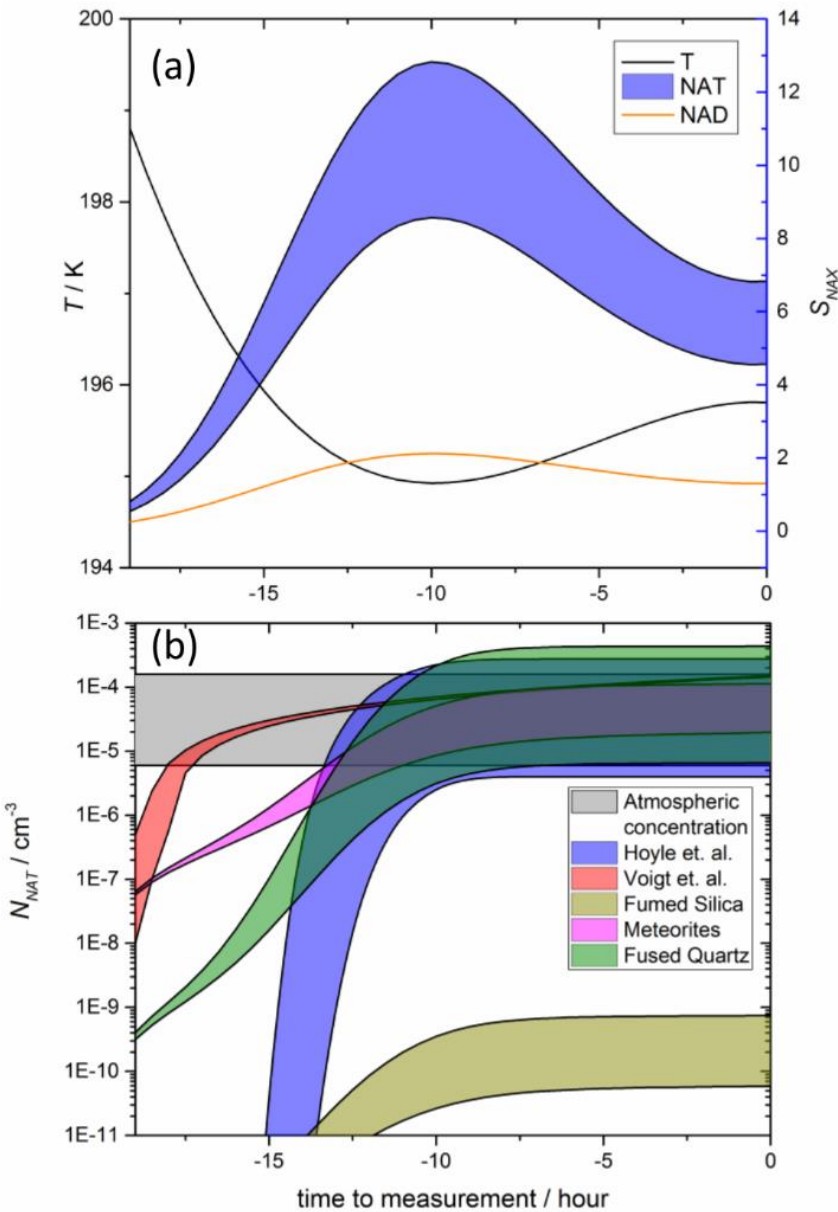

**Figure 6: NAT particle production using a temperature trajectory based on stratospheric observations. Temperature profile (a) was taken from back trajectories which ended in cloud with observed $N_{NAT}$ shown in the grey shaded area (b)(Voigt et al., 2005).** $S_{NAX}$ **shown in (a) were calculated at 70 hPa assuming 5 ppmv H$_2$O, 0.1 ppbv H$_2$SO$_4$ and 10 (minimum of ranges) to 15 ppbv (maximum of ranges) HNO$_3$. No other processes (e.g. growth or sedimentation of particles) are taken into account. (b) shows predicted $N_{NAT}$ based on our $n_s$ parameterisations and estimated surface areas of meteoric material as well as two literature parameterisations (Hoyle et al., 2013;Voigt et al., 2005).**