# Peer review of "Nucleation of nitric acid hydrates in Polar Stratospheric Clouds by meteoric material"

_Atmospheric Chemistry and Physics, 2017_

## Referee Comment (RC1) · Anonymous Referee #2 · 6 Nov 2017

**General Comments**

The manuscript presents an attracting study on the heterogeneous nucleation of crystalline NAT by meteoric materials analogues. The authors have shown that meteoric material can trigger nucleation in PSCs. The paper is well written and I recommend its publication in ACP.

**Specific Comments**

My main doubt is concerning to the different ability of the meteoritic material analogues tested to trigger the nucleation. The authors argue that olivine-pyroxene phase dominate the activity, but it looks that the specific surface area of the meteoric materials is also a key factor. According to the results shown, materials which present larger

specific surfaces areas show less ability to nucleate than those with much less surface area values (larger particles). My question is: what feature (specific surface area or olivine-pyroxene presence) is more important in this process?. In addition, these results sound strange to me, because usually nano-materials are much more efficient in heterogeneous processes as e.g. catalysis. The authors recognize that the explanation of these differences is currently not clear. Nevertheless, they argue that small particles are of similar order to the size of the critical clusters which is not good for nucleation and quote a paper of 1978, but I would like to see more details of such assumption. Also, they argue that relatively small changes in the surface properties of materials may have significant impacts on their nucleation activities. Which surface properties are important?, and also, it is possible to know something more about these relevant surface properties of the materials used? Also, when salts of $Fe^{3+}$ or $Mg^{2+}$ are added, no effect on the nucleating temperatures was observed, in contrast to the study of Wise et al, 2003. Some tentative explanation of this issue should be done. The authors also claim that NAT phase (instead of NAD) is formed directly during nucleation, but not any reference to which crystalline phase of NAT (alpha- or beta-) could be formed. Although, it is not possible to measure it in these experiments, I think that it is important to mention this issue. In fact, in a recent work (Weiss et al. Angew. Chem. Int. Ed. 2016, 55, 3276 –3280) it has been shown that the presence of alpha-NAT (instead of beta-NAT) could be the key step to explain the mechanism for NAT formation in high-altitude ice clouds. Although it is not the issue of this paper, the possible existence of different crystalline NAT phases and its relevance in the nucleation process should be mentioned in the paper.
* * *

---

## Referee Comment (RC2) · Anonymous Referee #1 · 12 Nov 2017

This short paper deals with a subject of current interest and reads well in its present concise version. It will in all likelihood find an interested community within the circle of readers of acp as the authors propose a novel and potentially important class of ice nucleation materials of extraterrestrial origin. However, the report gives a highly fragmentary view of the posed problem, whether or not meteoric material and its likely surrogates may make a significant contribution to NAT nucleation under suitable UT/LS conditions. The authors use a very limited data set (a single aqueous HNO3 concentration, no check for possible dependence of results w/r to cooling rate) associated with visual observation of the phase transition (liquid to solid and vice versa) leading to an unexplained result as far as fused quartz and fumed silica particles are concerned. The authors do not seem to attach great importance to the physical-chemical properties of the interface of the ice nucleating particles, instead they bring bulk properties into play (X-Ray diffraction and EDX) that are not informative or relevant in the present context. The reader is left with a snapshot of potentially interesting results without confidence that the authors are not prone to or victim of artefacts in the wake of the processing/handling of all tested materials except fumed silica, the only material used as is (I am assuming that the supplier of the quartz powder has ground the macroscopic sample using a ball mill as well!). In fact, according to Figure 4 all materials that have been ground to some extent except the "used as is" fumed silica samples (including "doped" ones) show a much larger nucleation activity than the latter at constant S(NAT). The authors should at least touch upon this fact in the discussion portion of the manuscript and include possible explanations. Please see below how one could at least in the case of fused silica particles make sure how to remove a potentially contaminated and/or amorphous interface by etching away a few molecular monolayers. I believe that the authors should include a more profound discussion as the present version is short on details and comes across as "superficial" (no pun intended!). Papers reporting limited aspects of nucleation like the present work abound throughout the literature, and the take-home lesson for the reader remains uncertain and unclear, if not confusing. The modeling part is informative but critically hinges upon the accuracy and veracity of the experimental results. It underlines the impact of the experimental results, if confirmed.

The following points/questions listed below need particular attention on the part of the authors. In my opinion it is necessary to include answers or explanations to every single point raised in the following in the revised version:

- A single aqueous 40%(wt) HNO3 concentration (corresponding to H2O-rich NAT) has been used, presumably to correspond to typical concentration/temperature conditions of HNO3 in the UT/LS of between 10 to 15 ppb (v/v). A burning question might be whether or not there is a HNO3 concentration range that could lead to NAT nucleation as well according to the Liquidus curve for NAT spanning roughly the HNO3

concentration range of 33 to 72%(wt) with the stoichiometric solution of 53.85%(wt) lying in-between (see for example Beyer and Hansen, J. Phys. Chem. A 2002, 106, 10275). There are measurements to the effect that strata of different atmospheric trace gas concentrations do exist in the UT/LS, among them for HNO3.

- Regarding the detection of the phase transition as well as the identification of the molecular composition of the binary mixture NAT/ice the manuscript leaves wanting. Considering first the identification of the condensate, the H2O-rich NAT binary mixture has an eutectic melting point of roughly 231 K and a final melting temperature of approximately 245 K (where the NAT surrounded in pure H2O is melting). As a complication, there is another eutectic melting point at 72% HNO3 between the Liquidus curves of NAT and NAM. It is not clear to this reviewer how the observation of melting by eye can distinguish between these two alternate nitric acid hydrates as the observation of the "melting" temperature is ambiguous in regards to identification of the molecular composition of the condensate. Which "melting" temperature was observed: eutectic or final? How was the new apparatus validated anyway (which nucleation system. Please disclose)? How did the authors, in the absence of any spectroscopic evidence, determine the HNO3/H2O condensate as NAT? It seems crucial to spend some time and effort at positively identifying the molecular composition of the condensate going beyond merely visual observation. An FTIR microsope might help in this regard, all the more so as absolute cross sections for alpha- and beta-NAT have been measured (Iannarelli and Rossi, JGR, 2016). (My guess is that beta-NAT is the relevant NAT modification in this work considering the temperature). On pg. 6 and 9 (third paragraph) the authors invoke their belief of "direct formation of NAT", that is shunting the formation of intermediate NAD: based on which observations?

- Regarding the freezing process the authors took a specific experimental protocol without spanning a range of time scales for the freezing process (First paragraph on pg. 4). In order to make a point I will briefly exaggerate: If there a single site leading to NAT nucleation on the meteoric material or its surrogate a single collision of HNO3

with this special site will lead to the construction of the NAT lattice if it "sticks" and does not desorb. However, this event may require a long time for it to happen, hence the importance of probing several (especially slow) temperature ramps. I could not find a single experimental run dedicated to this question. It is true that atmospheric time scales may never be duplicated in the laboratory, but I would like to see an effort in this direction!

- According to Figures 3 and 4 fumed silica offers ns values roughly four orders of magnitude lower than all other examined materials at S(NAT) = 10. As alluded to above this may occur because all high-nucleation rate materials have been mechanically ground, either by hand (pestle and mortar) or in a ball mill. Pure fused silica offers the unique advantage to wet-etch the possibly amorphous interface layer of ground fused silica using aqueous (concentrated) HF with our w/o H2O2 (so-called Piranha solution used routinely in the microelectronics industry to clean Si-wafers). In this way (after several washing cycles with ultraclean H2O) one may restore the "native" or natural fused silica interface. The challenge remains to explain the significant difference in the number of ns sites between fused and fumed silica despite the "normalization" to unit surface area using f(T) = 1 - exp(n(s) s). A serious method of interface characterization before performing nucleation experiments would seem in order, at least for meteoric surrogate material. "The apparent agreement between the active MSP analogue fused quartz and the meteoric samples is likely coincidental" (see pg. 6, second paragraph): Is it?

- I am unable to follow the author's explanations regarding the "morphology" as well as their differences on pg. 6, second paragraph. Please try again!

- The authors use "saturation" a lot. Please always indicate WHAT is saturated? Is saturation w/r to condensation of a vapour onto its own condensed phase (e.g. S(NAT)), or w/r to the site occupancy of ns (pg. 5, line 30)?

- Pg. 6, line 17: It is a binary system, therefore always NAT/ice at the used HNO3 concentration!!

- Pg. 8, line 27: Parametrization of what? Using which variables? Too vague!

---

## Author Comment (AC1) · 17 Jan 2018

The authors wish to thank both reviewers and several other colleagues, who communicated comments personally, for their contributions. In the following document, reviewers' reports have been split up into individual comments (shown in black) and responses are shown in blue.

**Report 1:**

This short paper deals with a subject of current interest and reads well in its present concise version. It will in all likelihood find an interested community within the circle of readers of acp as the authors propose a novel and potentially important class of ice nucleation materials of extraterrestrial origin. However, the report gives a highly fragmentary view of the posed problem, whether or not meteoric material and its likely surrogates may make a significant contribution to NAT nucleation under suitable UT/LS conditions.

We thank the reviewer for many insightful comments, which have led to an improved manuscript. We stress that the objective of this study was to see if materials of meteoric origin (fragments and smoke) have the capacity to nucleate nitric acid hydrates (not ice, as suggested by the referee); this has never previously been demonstrated in the laboratory. Our present study will motivate the study of the detailed understanding of why these materials nucleate nitric acid hydrates over a much broader range of atmospherically relevant polar stratospheric conditions (PSC are not normally thought of as a UT/LS phenomenon as suggested by the referee). We have therefor surveyed a range of materials which the state of the art literature suggests may be atmospherically relevant, and assessed which of these are sufficiently active to explain observed cloud crystal numbers.

This study is empirical in nature, but will form the basis of future work. This later work would aim to produce the level of detailed information required for predicting atmospheric behaviour. The reviewer in this case is advocating a very detailed examination of surface characteristics which might give a more fundamental understanding of the nucleation process. However, without the empirical approach we have employed here we would not know which, if any, meteoric materials have the capacity to trigger nucleation under atmospherically relevant conditions.

We have used a similar empirical approach for other atmospheric nucleation systems. For example, for ice nucleation in tropospheric clouds Atkinson et al. (2013) showed that K-feldspar likely controls mineral dust contributions to nucleation activity. Later Harrison et al. (2016) and Whale et al. (2017) used this work as the motivation and basis for investigations of material features which control the activity of K-feldspars revealing which mineral phases are particularly good at nucleating ice and also focusing in on surface features which play a role in nucleation.

We have therefore not implemented all of the suggested by this reviewer, but note that our paper will motivate significant future work, towards which these ideas may be very useful. Additions to the text have been made to reflect this discussion, and are detailed in response to individual comments.

The authors use a very limited data set (a single aqueous HNO3 concentration, no check for possible dependence of results w/r to cooling rate) associated with visual observation of the phase transition (liquid to solid and vice versa)

As stated above, our intention here was to survey a range of relevant materials to ascertain if these classes of materials have sufficient capacity to cause nucleation of nitric acid hydrates. With regard to changing $HNO_3$ concentration, we would point out that the saturation with respect to the condensed phase, which depends on the concentration and temperature, is the quantity which controls nucleation; so having accessed the atmospherically relevant regime we would not prioritise investigation of other $HNO_3$ binary solutions. We do agree, as stated in the text (page 6, second paragraph), that we have neglected the possible time dependence of nucleation, and also the effects of $H_2SO_4$ in Stratospheric Ternary Solution (STS) droplets. These would be our priorities for future investigation.

The discussion of time dependence in the results section has been expanded to suggest that this should be a topic of future research and now reads as follows.

"In this approach it is assumed that the number of nucleation events is primarily controlled by $S_x$ (which in turn is determined by temperature) of the system and that the time dependence of nucleation is of secondary importance. This approximation works well when the active sites across a surface are diverse in their ability to nucleate (Herbert et al., 2014). Testing the relative importance of time dependence of nucleation by the active materials identified here, using the methodology developed by (Herbert et al., 2014), would be a useful topic of future work."

leading to an unexplained result as far as fused quartz and fumed silica particles are concerned.

As mentioned above, the aim of this work was to identify potentially important materials, not to probe the mechanism of nucleation on any given material. The differences between these materials may provide a hint as to the mode of action, but we would regard that as a motivation for future work rather than necessary to the conclusions of this paper.

The following text has been added to the end of the introduction section to clarify the aims of this study.

"For each of these materials we aimed to determine their ability to nucleate nitric acid hydrates relevant to synoptic PSCs when immersed in binary $HNO_3$-$H_2O$ solutions. This survey across a range of proxies for meteoric material is designed to give an indication of whether MFs or MSPs can nucleate nitric acid hydrates efficiently enough to account for NAT particle observations in the polar stratosphere."

The authors do not seem to attach great importance to the physical-chemical properties of the interface of the ice nucleating particles, instead they bring bulk properties into play (X-Ray diffraction and EDX) that are not informative or relevant in the present context.

The bulk analyses we performed, reported here and also in James et al. (2017), were primarily for identifying the materials we were working with and, in the case of meteorites, confirming other reports already present in the literature. For example we excluded a third silica sample from the study because it was found to be crystalline. Bulk analyses are also relevant in that they allow us to discuss which phases are important for nucleation.

The following paragraph has been added to the methods section to clarify the purpose of analysis carried out here.

"Powder X-ray diffraction was used to confirm that the two $SiO_2$ materials contain no measurable crystalline component, as shown in the Supplementary Information (SI; see figure S1). Scanning Electron Microscopy with Energy Dispersive Electron spectroscopy (SEM-EDX, see figures S2 and S3) showed that the fused quartz consists of particles ranging from several hundred nm to several μm diameter, composed of Si and O only (detection limit of 0.1 %). Fumed silica was found to have a fractal morphology, made up primary particles of just ~6 nm, similar to MSPs and previous microscopy images of fumed silica (Bogdan et al., 2003)."

The reader is left with a snapshot of potentially interesting results without confidence that the authors are not prone to or victim of artefacts in the wake of the processing/handling of all tested materials except fumed silica, the only material used as is (I am assuming that the supplier of the quartz powder has ground the macroscopic sample using a ball mill as well!). In fact, according to Figure 4 all materials that have been ground to some extent except the "used as is" fumed silica samples (including "doped" ones) show a much larger nucleation activity than the latter at constant S(NAT). The authors should at least touch upon this fact in the discussion portion of the manuscript and include possible explanations.

In fact the $MgFeSiO_4$ synthesised in house was also ground with a pestle and mortar. The low activity of this material would seem to suggest that the grinding process does not influence the nucleation activity.

This has now been clarified in the methods section and discussed in the results section as follows.

"$MgFeSiO_4$ was synthesised from chemical precursors using a sol-gel method we have described previously (James et al., 2017), and ground on an agate pestle."

"The grinding process might introduce features to the surfaces which cause nucleation (Hiranuma et al., 2014), and indeed several of the more active samples were ground.  However, the $MgFeSiO_4$ sample was ground on a pestle and mortar but remains relatively inactive, hence the grinding process alone cannot account for the observed differences in activity."

Please see below how one could at least in the case of fused silica particles make sure how to remove a potentially contaminated and/or amorphous interface by etching away a few molecular monolayers.

See response to more detailed comment below.

I believe that the authors should include a more profound discussion as the present version is short on details and comes across as "superficial" (no pun intended!). Papers reporting limited aspects of nucleation like the present work abound throughout the literature, and the take-home lesson for the reader remains uncertain and unclear, if not confusing.

To reiterate: The assumption that nucleation is controlled by meteoric material has been made in a wide range of previous work (e.g. those papers summarised by Hoyle et al. (2013)) yet this is the first laboratory demonstration that that assumption may be reasonable. We present a survey of materials and a methodology to determine which are atmospherically relevant. The quantification of those activities to the level of detail required for atmospheric modelling will of course require further work, as will determining the details of the nucleation mechanism of any particular material, but neither of those was a stated aim of this paper. No changes have been made to the text as a direct result of this comment, though we hope that alterations made in response to other comments help to clarify the aim of our study.

The modeling part is informative but critically hinges upon the accuracy and veracity of the experimental results. It underlines the impact of the experimental results, if confirmed. The following points/questions listed below need particular attention on the part of the authors. In my opinion it is necessary to include answers or explanations to every single point raised in the following in the revised version:

- A single aqueous 40%(wt) HNO3 concentration (corresponding to H2O-rich NAT) has been used, presumably to correspond to typical concentration/temperature conditions of HNO3 in the UT/LS of between 10 to 15 ppb (v/v). A burning question might be whether or not there is a HNO3 concentration range that could lead to NAT nucleation as well according to the Liquidus curve for NAT spanning roughly the HNO3 concentration range of 33 to 72%(wt) with the stoichiometric solution of 53.85%(wt) lying in-between (see for example Beyer and Hansen, J. Phys. Chem. A 2002, 106, 10275). There are measurements to the effect that strata of different atmospheric trace gas concentrations do exist in the UT/LS, among them for HNO3.

This question has been the focus of significant work in the past (see for example Knopf et al. (2002)). Some studies have concluded that $HNO_3$ solutions more concentrated than those used here are possible, but generally only when gravity wave activity leads to rapid cooling and non-equilibrium aerosol (Meilinger et al., 1995). At concentrations significantly below 40 % we observed nucleation at temperatures where water ice is thermodynamically stable, whilst concentrations larger than 45 % are not thought to be relevant to these temperature conditions. Variation within those limits may affect nucleation, however other open questions, such as the impact of up to 5 % $H_2SO_4$ also need to be explored in future studies.

Covering significant portions of the ternary phase diagram would be possible with the current technique and indeed is a key aim of future work. However, carrying this out for the range of materials here would be unreasonably time consuming. The first step must be to identify the most important materials, then to assess their behaviour under a wide range of atmospherically relevant conditions to facilitate robust atmospheric modelling of heterogeneous nucleation of PSCs.

The description of the $HNO_3$ concentrations selected in the methods section has been expanded as below. Discussion of the importance of $H_2SO_4$ is also present throughout the manuscript.

"$HNO_3$ concentrations higher than about 45 wt% are not thought to be relevant to the atmosphere in the absence of gravity wave-induced temperature perturbations (Meilinger et al., 1995), whilst lower concentrations lead in this experimental setup to nucleation under conditions where water ice is stable, also not relevant to the synoptic clouds we focus on here (Mann et al., 2005)."

- Regarding the detection of the phase transition as well as the identification of the molecular composition of the binary mixture NAT/ice the manuscript leaves wanting. Considering first the identification of the condensate, the H2O-rich NAT binary mixture has an eutectic melting point of roughly 231 K and a final melting temperature of approximately 245 K (where the NAT surrounded in pure H2O is melting). As a complication, there is another eutectic melting point at 72% HNO3 between the Liquidus curves of NAT and NAM. It is not clear to this reviewer how the observation of melting by eye can distinguish between these two alternate nitric acid hydrates as the observation of the "melting" temperature is ambiguous in regards to identification of the molecular composition of the condensate. Which "melting" temperature was observed: eutectic or final?

The description of melting in the methods section has been expanded as below to indicate that the onset of melting was observed at the eutectic temperature, whilst the gradual melting made visual determination of the final melting point impossible.

"Phase transitions on warming, which tend to occur without any kinetic limitation, can be a useful way of both validating the reported temperatures and identifying the phases which had crystallised. On warming, simultaneous visual changes in all the frozen droplets started at 231.2 ± 0.7 K and continued until no visible solid was left in the droplets. The temperature at which changes start in the droplet is consistent with the ice $I_h$ / NAT eutectic at 231 K (Martin, 2000) followed by the gradual melting of NAT in equilibrium with a binary $HNO_3$-$H_2O$ solution. Unfortunately, the temperature at which the NAT completely disappeared from the droplets, which would correspond to the NAT-aqueous solution line, was not possible to determine with any confidence due to the optical resolution of the camera, and therefore we do not report it. Nevertheless, the general behaviour is consistent with the established $HNO_3$–$H_2O$ phase diagram (Beyer and Hansen, 2002); this provides further confidence in our reported temperatures."

How was the new apparatus validated anyway (which nucleation system. Please disclose)?

The system has been previously validated by melting points of three organic compounds and $H_2O$ ice (see Table 1 of Whale et al. (2015)), and here by observation of the melting onset at the eutectic temperature and by comparison of the internal temperature measurement to an external, calibrated PT100 thermocouple, which agreed to within 1 K at the base temperature of 183 K (as already stated in the text). The system has also been validated against other nucleation instruments by Hiranuma et al. (2015).

The description at the beginning of the methods section has been expanded and now reads as follows.

"Drop freeze assays were performed using a modified version of the Nucleation by Immersed Particles Instrument (NIPI), which we have described previously and has primarily been used to study heterogeneous ice nucleation (Whale et al., 2015). Generally, this technique involves generating aqueous suspensions of particles and then pipetting an array of droplets of known volume onto an appropriate surface on a cold stage. They are then cooled and the freezing point determined optically. The NIPI system has previously been validated by observing well-defined melting points of a variety of organic compounds and ice (Whale et al., 2015). In addition, results from this instrument also compare favourably to a range of other droplet-based techniques for measuring the nucleation efficiency of immersed particles (Hiranuma et al., 2015)."

How did the authors, in the absence of any spectroscopic evidence, determine the HNO3/H2O condensate as NAT?

See following response to both comments.

On pg. 6 and 9 (third paragraph) the authors invoke their belief of "direct formation of NAT", that is shunting the formation of intermediate NAD: based on which observations?

Our statements on the formation of NAT are not based on 'belief'. The main determination that NAT nucleated directly was that some experiments showed nucleation under conditions where no other phases were thermodynamically stable. This was the case for all nucleation events observed when using fused quartz, and some for the meteorite samples. In the case of all events with the less active $MgFeSiO_4$ and fumed silica and the majority of events with meteorites, NAD was also thermodynamically stable and therefore could possibly have been the nucleating phase. Since there is no discontinuity in the trend of $n_s$ with $S_{NAT}$ at the point where NAD becomes stable, we have made the simplifying assumption that all nucleation led directly to NAT.

The results section has been expanded to explicitly state that the observation of nucleation above $T_{NAD}$ can only be explained by direct formation of NAT since there are no other thermodynamically stable solids. No further changes have been made to the text since the possible presence of metastable NAD in some cases was already explicitly stated and this reasoning for the identification of NAT was already included in the first and last paragraphs of the results section.

It seems crucial to spend some time and effort at positively identifying the molecular composition of the condensate going beyond merely visual observation. An FTIR microsope might help in this regard, all the more so as absolute cross sections for alpha- and beta-NAT have been measured (Iannarelli and Rossi, JGR, 2016). (My guess is that beta-NAT is the relevant NAT modification in this work considering the temperature).

This may be an interesting topic for future work, however the bulk phase observed may not be that which initially nucleated. The phase which results after nucleation and crystal growth depends on details of the crystal growth. If a metastable phase initially nucleates, this can relax to a more stable phase during crystal growth. Hence, identifying the phase which results from crystallisation does not necessarily tell you about the phase which nucleates.

The final paragraph of the results section has been expanded to include this discussion:

"Some metastable NAD may form when SNAD > 1, however the consistent melting onset of droplets in agreement with the NAT / ice eutectic suggests that if any NAD did form it converted to NAT. We did not attempt to identify the phase of the acid hydrate in the frozen droplets, since the polymorph resulting from crystallisation may not be the same phase which initially nucleated. If a metastable phase nucleates, it often converts to a more stable phase during the crystallisation process (Murray and Bertram, 2008)."

- Regarding the freezing process the authors took a specific experimental protocol without spanning a range of time scales for the freezing process (First paragraph on pg. 4). In order to make a point I will briefly exaggerate: If there a single site leading to NAT nucleation on the meteoric material or its surrogate a single collision of HNO3 with this special site will lead to the construction of the NAT lattice if it "sticks" and does not desorb. However, this event may require a long time for it to happen, hence the importance of probing several (especially slow) temperature ramps. I could not find a single experimental run dedicated to this question. It is true that atmospheric time scales may never be duplicated in the laboratory, but I would like to see an effort in this direction!

As discussed above, time dependence is a potentially important aspect of the nucleation activity of materials, and one which we would expect to investigate before

presenting a model for robust analysis of observed atmospheric nucleation. Here we identify materials which may be important by examining the limiting case with no time dependence. Note that if time dependence is found to be significant, this will only increase the predicted activity on atmospheric timescales, not altering the conclusions for materials which were found to be sufficiently active.

A statement that time dependence should be a topic of future research has been added to the results section.

- According to Figures 3 and 4 fumed silica offers ns values roughly four orders of magnitude lower than all other examined materials at S(NAT) = 10. As alluded to above this may occur because all high-nucleation rate materials have been mechanically ground, either by hand (pestle and mortar) or in a ball mill. Pure fused silica offers the unique advantage to wet-etch the possibly amorphous interface layer of ground fused silica using aqueous (concentrated) HF with our w/o H2O2 (so-called Piranha solution used routinely in the microelectronics industry to clean Si-wafers). In this way (after several washing cycles with ultraclean H2O) one may restore the "native" or natural fused silica interface.

We argue that removing surface layers from these silica materials by chemical processing e.g. with HF would not result in more atmospherically relevant material. This may be a useful experiment in determining characteristics which control the nucleation activity of silica, however since that is not an aim of the present work we have not included discussion of this possibility in the text.

The challenge remains to explain the significant difference in the number of ns sites between fused and fumed silica despite the "normalization" to unit surface area using f(T) = 1 - exp(n(s) s). A serious method of interface characterization before performing nucleation experiments would seem in order, at least for meteoric surrogate material.

This is where our preferred approach differs to that of the reviewer. Taking the assumption that relatively rare sites on the surface control nucleation ability of the material, detailed characterisation of the surface may or may not identify the characteristic of the material which controls its nucleation activity. Our approach, which focusses on atmospheric relevance, is to identify the important materials first, then characterise the most important ones in further work. Thus this study is analogous to the study of Atkinson et al. (2013), which identified K-feldspar as important for ice nucleation in mixed phase clouds, whilst future work would be analogous to the publications of Whale et al. (2017), which identified possible characteristics of that material which might be critical to the nucleating activity.

The aims of the present study have been clarified in the final paragraph of the introduction section, as reproduced above.

"The apparent agreement between the active MSP analogue fused quartz and the meteoric samples is likely coincidental" (see pg. 6, second paragraph): Is it?

The language in the text has been softened to "may be coincidental", however the lack of any significant silica phase in meteorites makes it unlikely that the same material characteristic controls nucleation in these cases.

- I am unable to follow the author's explanations regarding the "morphology" as well as their differences on pg. 6, second paragraph. Please try again!

This discussion has been reworded, particularly in response to comments in the second reviewer's report, see reproduced text below.

- The authors use "saturation" a lot. Please always indicate WHAT is saturated? Is saturation w/r to condensation of a vapour onto its own condensed phase (e.g. S(NAT)), or w/r to the site occupancy of ns (pg. 5, line 30)?

Saturation has been defined symbolically on its first use as the reviewer suggests (using a capital $S$ with a subscript to denote the phase(s) in question) and altered throughout the text. The specific line mentioned has also been reworded to clarify that $n_s$ is a function of $S_x$.

- Pg. 6, line 17: It is a binary system, therefore always NAT/ice at the used HNO3 concentration!!

The relevant line has been edited to include the presence of $H_2O$ ice.

- Pg. 8, line 27: Parametrization of what? Using which variables? Too vague!

$n_s(S_{NAT})$ has been used to clarify that the parametrisation referred to is that produced in the current study.

**Report 2:**

The manuscript presents an attracting study on the heterogeneous nucleation of crystalline NAT by meteoric materials analogues. The authors have shown that meteoric material can trigger nucleation in PSCs.
The paper is well written and I recommend its publication in ACP.

We are pleased the referee is generally supportive.

My main doubt is concerning to the different ability of the different meteoritic material analogues tested to trigger the nucleation. They argue that olivine-pyroxene phase dominate the activity, but it looks that the specific surface area is also a key factor. According to the results shown, materials which present larger specific surfaces areas shown less ability than that with much less surface area values (larger particles). My question is: what feature (specific surface area or olivine-pyroxene presence) is more important in this process?

Both the presence of materials which have an inherent ability to trigger nucleation and the size of the particles of that material are important in heterogeneous nucleation.

If a particle is small it will nucleate less effectively than a larger particle because it has a lower probability of containing an active site; this is accounted for in the term $n_s$. But, theoretically, there is an additional factor which reduces the nucleation efficiency (expressed as $n_s$) for small particles: as their size approaches the size of the critical cluster of nitric acid hydrate, they are less capable of stabilising the crystalline cluster and therefore are thought to be less effective at nucleation. This may be an issue with some of the nano-scaled particles used here.

Also note that we suggest that the olivine-pyroxene phase is important for the ground meteorites, it obviously cannot be important for the silica samples.

In response to this and a comment from the first reviewer the discussion in the results section has been reworded significantly, giving an itemised discussion of possible properties which may control nucleation in this case, including the following.

"Alternatively, the significantly smaller activity of the fumed silica and $MgFeSiO_4$ may be a result of their morphology, specifically the particle size. Both have specific surface areas around 200 $m^2$ $g^{-1}$, corresponding to an equivalent spherical radius of ~6 nm (Bogdan et al., 2003;James et al., 2017).  Such small particles are of a similar order to the size of critical clusters and according to classical nucleation theory are therefore thought to be relatively poor at causing nucleation (Pruppacher and Klett, 1978)."

In addition, the results sound strange to me, because usually nano-materials are much more efficient in heterogeneous processes as e.g. catalysis. The authors recognize that the explanation of these differences is currently not clear. Nevertheless, they argue that small particles are of similar order to the size of the critical clusters which is not good for nucleation and quote a paper of 1978, but I would like to see more details of such assumption.

Nano materials can be more effective in catalysis of chemical reactions because they have a high surface area.  But, in nucleation, the particle size is important as well as the inherent ability of that material to trigger nucleation. This is because in e.g. gas phase catalysis, only a few molecules must come together to react whereas in the case of nucleation, critical clusters must form from many molecules and can then be of a nanometer scale. If the particles are of a similar or smaller size to the critical clusters they are thought to be less effective at stabilising the cluster.

On the issue of particle size vs. critical cluster size, the reference from 1978 is a standard text book (Microphysics of Clouds and Precipitation) and we do not want to reproduce a detailed discussion of why small particles are thought to be less efficient at nucleating ice according to classical nucleation theory. But, we have adjusted the pertinent sentence to make this concept clearer (reproduced above).

Also, they argue that relatively small changes in the surface properties of materials may have significant impacts on their nucleation activities. Which surface properties are important? and also, it is possible to know something more about these relevant surface properties of the materials used?

See responses to first reviewer comments, particularly as regards the analogy to the K-feldspar work.

Also, when salts of Fe3+ or Mg2+ are added, no effect on the nucleating temperatures was observed, in contrast to the study of Wise et al, 2003. Some tentative explanation of this issue should be done.

The following discussion of the differences between our study and that of (Wise et al., 2003) has been added to the results section.

"The authors in that case speculated that soluble $Fe^{3+}$ or a combination of that with other metal ions affected the nucleation process. The lack of a similar effect here could be a result of working in a different acid solution, nucleating a different phase or the differing volume of samples. That study also differs from this in that our experiments include particles which control the nucleation and may have active sites which are not susceptible to the chemical effects of the dissolved metals."

The authors also claim that NAT phase (instead of NAD) is formed directly during nucleation, but not any reference to which crystalline phase of NAT (alpha- or beta-) could be formed. Although, it is not possible to measure it in these experiments, I think that it is important to mention this issue. In fact, in a recent work (Weiss et al. Angew. Chem. Int. Ed. 2016, 55, 3276 –3280) it has been shown that the presence of alpha-NAT (instead of beta-NAT) could be the key step to explain the mechanism for NAT formation in high-altitude ice clouds. Although it is not the issue of this paper, the possible existence of different crystalline NAT phases and its relevance in the nucleation process should be mentioned in the paper.

The following discussion of the wider literature on NAX polymorphs has been added to the introduction.

"Several authors have examined the homogeneous nucleation of nitric acid hydrates and it has been noted that nucleation may occur via metastable Nitric Acid Dihydrate (NAD, α- or β- polymorphs (Grothe et al., 2008)) or α-NAT (Weiss et al., 2016), which later transform to the stable β-NAT. However, it is thought that the homogeneous nucleation of nitric acid hydrates, either through surface or volume pathways, is not sufficiently rapid to cause observed NAT crystal concentrations in the atmosphere (Knopf et al., 2002; Knopf, 2006; Stetzer et al., 2006; Möhler et al., 2006)."

The following discussion of the possible polymorph formed in this study and in the atmosphere has also been added to the results section.

"While observations indicate that NAT is the phase which exists in PSC (Höpfner et al., 2006), it is possible that other metastable nitric acid hydrate phases (Nitric Acid Dihydrate, α- or β-NAD) may form initially, then transform to the stable NAT phase (Grothe et al., 2008;Weiss et al., 2016). We note that the 820 cm$^{-1}$ feature used by Höpfner et al. (2006) to identify atmospheric NAT is present for both the α- and β- polymorphs (Iannarelli and Rossi, 2015). Since the equivalent 816 cm$^{-1}$ feature for β-NAD has not to our knowledge been compared to the atmospheric spectra there is still uncertainty regarding the relevant atmospheric phases. In fact, NAD nucleation has been observed under atmospheric conditions for homogeneous nucleation (Stetzer et al., 2006)."

Citations

Atkinson, J. D., Murray, B. J., Woodhouse, M. T., Whale, T. F., Baustian, K. J., Carslaw, K. S., Dobbie, S., O'Sullivan, D., and Malkin, T. L.: The importance of feldspar for ice nucleation by mineral dust in mixed-phase clouds, Nature, 498, 355, 10.1038/nature12278, 2013.

[revised manuscript text omitted]

---

## Referee Report (RR1)

**Referee Report on manuscript no. acp-2017-816 « Nucleation of nitric acid hydrates in Polar Stratospheric Clouds by meteoric material » by A.D. James, J.S.A. Brooke, T.P. Mangan, T.F. Whale, J.M.C. Plane and B.J. Murray submitted to Atmospheric Chemistry and Physics**

I acknowledge the effort of the authors to amend the manuscript in the sense suggested by both referees. However, I think that the problems lie at a deeper level where cosmetic changes to the text just won't do. We are dealing with a manuscript that examines ocular (by eye) freezing and melting observations of a single known $HNO_3$ concentration (40%wt) as a function of temperature in the presence of processed meteoric material and some of its analogues. The identity of the nucleating phase remains unexamined, and conjectures on the presence of the stable phase are made after thermodynamic relaxation according to published phase diagrams. In a stricter sense the chosen title is misleading because nucleation is never investigated.

In the following I would like to raise a few significant points:

Regarding the use of a single $HNO_3$ solution I point out that known equilibrium phase diagrams are NOT a good guide to interpret observations on nucleation which is a kinetic process. Rather, a cascade from one or more metastable states towards a final thermodynamically stable state is observed according to Ostwald's rule. The formation of NAD ($\alpha$ and $\beta$-modification) via homogeneous nucleation of a stoichiometric gas mixture ($HNO_3:H_2O$ =1:3) in the AIDA chamber (Stetzer 2006) and metastable $\alpha$-NAT via heterogeneous immersion and deposition freezing (Grothe 2008, Weiss 2016) are examples. Given the fact that the authors intend to examine the nucleation, a kinetic process, I think that a concentration range from 33 to 45%wt would be more appropriate in order to free themselves from the narrow range given by equilibrium considerations that certainly kick in later in the process upon thermodynamic relaxation. The authors report the onset (eutectic) melting at 231.2 K, however, the phase diagram (Beyer, 2002) reveals an eutectic tie line at the same temperature in the range 0 to 78%wt spanning the range all the way from ice, NAT, NAD and NAM. This makes the observation of the onset of melting ambiguous in nature.

In addition, at 40%wt of $HNO_3$ these workers spot an unknown hydrate having a peritectic point (incongruent melting) that they claim is unimportant at equilibrium, but which cannot be ruled out as a nucleation phase. Soleley based on that fact it is absolutely unjustified to conclude that NAT is nucleated "directly" without the implication of metastable phases that have been found in both homogeneous and heterogeneous nucleation experiments. The present underdiagnosed laboratory observations do not provide a basis for the statement of spontaneous nucleation of NAT (see pg. 8, line 2: pg. 10, line 25), all the more so that there are numerous potential metastable states whose eutectic melting starts at 231 K (Beyer 2002). The chosen laboratory methods in this work are just too crude in order to sort out what is going on upon nucleation of an aqueous $HNO_3$ solution.

Regarding the abscissa of Figure 4 I do not understand the insistence of the authors to codify the temperature in terms of saturation ratios of NAT or NAD. The temperature is the independent variable in this case and should be used in Figure 4. The equilibrium phase diagram is irrelevant here and is grossly misleading the reader by suggesting that nucleation is an equilibrium phenomenon. Saturation of a NAT or NAD phase has nothing to do with the process at hand which will depend on the adsorption of $HNO_3$ and $H_2O$ vapor to the interface of meteoric materials and their analogues (proxies). Hoyle (2013) has used the "theory of active sites" successfully which is nothing else than a simple Langmuir-Hinshelwood model that has to be extended to allow for multilayer adsorption!

The authors are dismissing a bit too rapidly my original observation that the interface of the refractory samples may have been modified (rendered amorphous) in the course of the grinding process. In a stricter sense all the examined meteoric materials as well as the "analogues" are atmospherically irrelevant because they may have undergone changes in the grinding process. Sometimes this manifests itself in surface-sensitive diffraction processes where one observes line-broadening (for instance in small particles or powders offering large surface-to-bulk ratios). I must admit that the analogue $MgFeSiO_4$ does not fit that scheme as it still is fairly inactive! It is possible that the argument is more complicated and that grinding is perhaps not the only salient parameter! However, to take the single example of the above olivine material as a reason to dismiss the argument seems superficial and not justified. The one and only material that is not processed in fumed silica and its doped congeners (displayed in the first column on top of Figure 4).

In the end I would like to emphasize a point made by Grothe (2008) and reiterated by Weiss (2016) that the reason for the transient stability (metastability) of NAD or $\alpha$-NAT is the stabilization afforded by the presence of water ice. DFT calculations in Weiss (2016) have quantified the interaction of $\alpha$-NAT with water ice and have found it to be larger than for $\beta$-NAT which itself is more stable than $\alpha$-NAT. There is a delicate balance of energetic terms on the way of a metastable phase to its relaxed (stable) final phase in view of the fact that the implied energy differences are small, albeit significant at the low temperatures of interest. Along these lines Iannarelli (2016) have examined the spontaneous formation of NAT and NAD on thin water ice films and have found consistently that $\alpha$-NAT is always formed before phase transition to the stable final phase that starts in the range 190 to 195 K. A few degrees may make a large difference in the kinetic stability of a metastable phase.

Regarding the introduction I would like to emphasize that both Grothe (2008) and Weiss (2016) dealt with heterogeneous nucleation of $HNO_3$-hydrates, reportedly in the immersion or contact freezing mode (liquid $N_2$) as well as in deposition freezing (cold metal support). The authors make it sound as if homogeneous nucleation was involved (pg. 2, line 2). I will not dwell on this "classical" characterization (Pruppacher) as these terms are devoid of any meaning in a mechanistic sense.

References are missing: Peter and Gross (2012); Lambert et al. (2016).

In conclusion, my problem with the present work is that the claims are not supported by established facts as well as experimental observations by the authors that are too "pedestrian" to deduce mechanistic details on nucleation of $HNO_3$ hydrates on meteoric materials and their analogues. The experimental material at hand is interesting though, but does not make a "story" at this stage owing to its fragmentary nature. The modeling part is OK if it were not "punished" by the rash experimental conclusions. The absent proof for the existence of an (expected) effect cannot be construed as an argument in favor of its contrary! Science is deductive by nature and cannot accept negative differential diagnostics ("This must be the reason by exclusion because I cannot think of a positive fact supporting my hypothesis"). There is too much speculation owing to the absence of experimental evidence!

---

## Author Response (AR2)

**Response to Referee Report on manuscript no. acp-2017-816 « Nucleation of nitric acid hydrates in Polar Stratospheric Clouds by meteoric material » by A.D. James, J.S.A. Brooke, T.P. Mangan, T.F. Whale, J.M.C. Plane and B.J. Murray submitted to Atmospheric Chemistry and Physics**

Dear Editor,

Thank you for your time and understanding during the review process. The authors wish to reiterate that the paper as it stands presents the first quantitative demonstration that meteoric material is a sufficiently active heterogeneous nucleus to explain atmospherically observed PSC (which has been hypothesised in a large number of studies over the last 3 decades). In addition we have identified meteoric fragments, a class of aerosol previously unrecognised in the stratosphere, as an alternative to meteoric smoke in this heterogeneous nucleation. The findings, methodology and techniques presented in this study could pave the way for answering atmospherically relevant questions which have been outstanding for several decades.

Of the two reviewers, one was willing to see the paper published, with only minor reservations. The second reviewer was rather more critical. Some of the comments from this second reviewer will lead to clarification and overall improvement of the text, and we are happy to implement these. However, other suggestions are not practicable: we are not able to satisfy the reviewer's request for further experiments since they are either not relevant to our aims, not achievable within a reasonable timescale (the CODITA grant which funded this research ended in March 2017) and in some cases, not achievable by any experimental technique which we are aware of. Here we present a generalised response to many of the reviewer's comments, followed by specific responses to those which have been addressed in the revised manuscript.

- The reviewer repeatedly states that referring to the equilibrium phase diagram and the use of saturation ratios is inappropriate. We are disappointed to see such a fundamental criticism being brought up so late in the review process. In fact, vast quantities of data are available from multiple chemical fields which show that increasing saturation ratio increases the probability of nucleation events occurring. The formulation of classical nucleation theory is entirely based on this premise and we do not know of any published literature that disputes the idea that the probability of nucleation increases with increasing supersaturation, in any system. Saturation (the ratio of free energy of the metastable liquid phase to the solid phase, as commonly defined in immersion mode nucleation) is recognised throughout the heterogeneous nucleation community as the variable which controls the probability of nucleation [e.g. (Koop et al., 2000)]. Hence we consider the use of saturation, based on phase diagrams, to be a reasonable way to parameterise the nucleation process.
- On the identity of the initially nucleating phase: in the previous version we stated that it is "likely" that NAT is the primary nucleating phase. We then made the assumption that this is so in order to parameterise our data and compare to atmospheric observations. We have edited the text (see below), stating that a different nitric acid hydrate may form first and clarifying that formation of NAT is a necessary assumption to allow atmospheric implications of the study to be probed. However we are not aware of any technique which can experimentally identify the critical cluster, the solid phase at the point of nucleation, since this is both very small (nm radius) and very short lived. This is an issue well understood in a range of nucleation communities. As a measure of the sensitivity of the atmospheric

implications to this assumption we have added a statement that if NAD is assumed to nucleate first, we see several orders of magnitude more solid particles in our atmospheric prediction. To assume that a different phase formed first we would have to guess at the thermodynamic properties of that phase and program these into the e-AIM model, a very time consuming task with little scientific basis.

It also seems that the reviewer is conflating our discussion of the observed melting with our identification of the phase. The purpose of including that observation in the paper is the confidence which it gives in the temperature measurement and control. Since the two eutectic points occur within such a small range of temperature, this is not affected by the identity of the phases melting. We have added (see below) a statement explicitly stating that our tentative identification of the nucleating phase does not rely on the observed melting point.

- A number of comments suggest that the reviewer continues to misunderstand the aims of our study:

  The reference to the "stabilization afforded by the presence of water ice" shows that the reviewer continues to misunderstand our clearly stated aim of studying clouds which form in the absence of water ice (at least 20% of denitrification [ref. Mann]). This also links to the question of performing experiments over a range of concentrations, where we already explicitly state that the choice of concentration is designed to facilitate identification of the phase which forms by excluding the possibility that water ice initially nucleates.

  Regarding alteration of the surfaces of particles during grinding: our explicitly stated aim is to survey a range of materials to assess which may be active enough to cause nucleation in the atmosphere (which we have achieved). Examining the exact mode of action of those materials, toward which these studies would indeed be valuable, is explicitly stated as a subject of future study. We are also puzzled by the continued references to "rendering amorphous" materials which are already amorphous in the bulk. This is also evident in the statement that our experimental observations are "too pedestrian to deduce details of the mechanistic details on nucleation". We might not choose this experiment to answer that question, but it is explicitly not the question we are trying to answer! It is worth noting that in the much more thoroughly studied system of water-ice nucleation such mechanistic details remain a very open question (Slater et al., 2016).

We now present responses (in blue font) to the majority of the reviewer's comments (in black font) along with details of changes made to the paper. Comments which have been generally responded to above and did not result in changes to the text have been omitted.

**Reviewer 1**

I acknowledge the effort of the authors to amend the manuscript in the sense suggested by both referees.

We are pleased that the reviewer appreciates the significant work which went into this review process.

We are dealing with a manuscript that examines ocular (by eye) freezing and melting observations of a single known $HNO_3$ concentration (40 %wt) as a function of temperature in the presence of processed meteoric material and some of its analogues. The identity of the nucleating phase remains unexamined, and conjectures on the presence of the stable phase are made after thermodynamic relaxation according to published phase diagrams.

The language in the text (final paragraph of Section 3) has been altered to clarify that the assumption of NAT as the primary phase which nucleates is made in order to allow investigation of atmospheric implications. We have also expanded the discussion (same paragraph) on how reasonable this assumption is. A statement has also been added (penultimate paragraph of Section 4.2) describing the effect of assuming that NAD nucleates first. The unchanged conclusions of the study suggest that the model is reasonably insensitive to the assumption of primary nucleating phase, which increases confidence that our conclusions are robust. We have adjusted the conclusions section to make it clear that we 'suggest' that NAT nucleates.

The altered paragraphs read as follows:

Section 3:
"In order to assess the atmospheric implications of these observations, the assumption has been made that the nucleation events observed in this study were direct nucleation of NAT. While observations indicate that NAT is the phase which exists in PSC (Höpfner et al., 2006), it is possible that other metastable nitric acid hydrate phases (Nitric Acid Dihydrate, α- or β-NAD) may form initially, then transform to the stable NAT phase (Grothe et al., 2008;Weiss et al., 2016). We note that the 820 cm$^{-1}$ feature used by Höpfner et al. (2006) to identify atmospheric NAT is present for both the α- and β- polymorphs (Iannarelli and Rossi, 2015). Since the equivalent 816 cm$^{-1}$ feature for β-NAD has not to our knowledge been compared to the atmospheric spectra there is still uncertainty regarding the relevant atmospheric phases. In fact, NAD nucleation has been observed under atmospheric conditions for homogeneous nucleation (Stetzer et al., 2006). However, in our experiments some nucleation events were observed under conditions where NAD is not thermodynamically stable ($S_{NAD}$ < 1), and since there is no significant discontinuity in the trend in $n_s$ at $S_{NAD}$ = 1, the assumption of direct nucleation of NAT seems reasonable. Since it is possible that a different nitric acid hydrate phase formed in these experiments we have examined the sensitivity of our atmospheric conclusions to the assumption of NAT as the primary nucleating phase. Some metastable NAD may form when $S_{NAD}$ > 1 (or some other metastable nitric acid phase may form); however, the consistent melting onset of droplets in agreement with the NAT / ice or NAT / NAM eutectic suggests that if any NAD did form it converted to NAT (note that the melting point is not taken to be supporting evidence of which phase nucleated initially). We did not attempt to identify directly the phase of the acid hydrate in the frozen droplets, since the polymorph resulting from crystallisation may not be the same phase which initially nucleated. In fact, if a metastable phase nucleates, it often converts to a more stable phase during the crystallisation process (Murray and Bertram, 2008). The parameterisations of $n_s$ as a function of $S_{NAT}$ shown in Figure 4 were therefore used to investigate the activity of meteoric material in heterogeneously nucleating PSC formation in the atmosphere."

Section 4.2:

"To test the sensitivity of the system to the assumption that NAT was the primary nucleating phase, parameterisations of $n_s$ as a function of $S_{NAD}$ were produced for the meteorite samples and the fumed silica. Note that a parameterisation was not produced for the fused quartz since with this material heterogeneous nucleation was always observed under conditions too warm (by up to 10 K) for NAD to be thermodynamically stable. These parameterisations were then implemented in the same atmospheric scenarios (temperatures and concentrations). The result was that the onset of nucleation was predicted 1.2-1.7 K colder, but that by the point of measurement around 250 times higher $N_{NAX}$ was predicted (data not shown). This means that the conclusions remain the same for both the meteorites (which are sufficiently active to explain observed cloud) and fumed silica (which is not), suggesting that the atmospheric conclusions of this study are reasonably insensitive to the choice of primary nucleating phase."

Conclusions:

 "Heterogeneous nucleation by analogues for Meteoric Smoke Particles (MSPs) and Meteoric Fragments (MFs) in binary HNO3 / H2O solutions has been measured in the laboratory. Both MSPs and MFs immersed in nitric acid solution droplets were found to nucleate crystalline nitric acid hydrates.  Given nucleation occurred under conditions where the metastable nitric acid dihydrate (NAD) is unstable, we suggest that the nitric acid trihydrate (NAT) initially nucleated, although we cannot rule out the nucleation of other metastable phases. Parameterisations of the resulting activity were used to show that heterogeneous nucleation on meteoric material is a potential pathway to forming observed number densities of NAT crystals in Polar Stratospheric Clouds (PSCs) in the absence of water ice."

In a stricter sense the chosen title is misleading because nucleation is never investigated.

It is true that we do not directly observe nucleation, we observe the result of nucleation – i.e. crystallisation.  It is a very common approach to take the crystallisation of a droplet as evidence of nucleation since crystallisation could not occur without nucleation.  Hence, we intend to keep the title unchanged.

In the following I would like to raise a few significant points:
Regarding the use of a single $HNO_3$ solution I point out that known equilibrium phase diagrams are NOT a good guide to interpret observations on nucleation which is a kinetic process.
We discuss this point in the first of this response. We think that the changes mentioned above clarify these issues.

The authors report the onset (eutectic) melting at 231.2 K, however, the phase diagram (Beyer, 2002) reveals an eutectic tie line at the same temperature in the range 0 to 78%wt spanning the range all the way from ice, NAT, NAD and NAM. This makes the observation of the onset of melting ambiguous in nature.

The observation of the melting point is included in the text primarily to give confidence in the temperature measurement and control of the apparatus, which would still be relevant in the case of either eutectic point. The text in Section 2 has been revised to state that "The temperature at which changes start in the droplet is consistent with either the ice $I_h$ / NAT or NAT / NAM eutectic". See also changes above regarding changes to the final paragraph of Section 3.

Saturation of a NAT or NAD phase has nothing to do with the process at hand which will depend on the adsorption of $HNO_3$ and $H_2O$ vapor to the interface of meteoric materials and their analogues (proxies).

Saturation ratios are entirely appropriate here and are not limited to defining gas phase adsorption. We stress that we are dealing with nucleation from a supersaturated solution, this is very clear in the manuscript.

Hoyle (2013) has used the "theory of active sites" successfully which is nothing else than a simple Langmuir-Hinshelwood model that has to be extended to allow for multilayer adsorption!
And

I will not dwell on this "classical" characterization (Pruppacher) as these terms are devoid of any meaning in a mechanistic sense.

These comment seems to be levelled more broadly at the extensive community that use these concepts. This paper is not the right place to open up this discussion. We have not made any changes in light of these comments.

In the end I would like to emphasize a point made by Grothe (2008) and reiterated by Weiss (2016) that the reason for the transient stability (metastability) of NAD or ☐-NAT is the stabilization afforded by the presence of water ice. DFT calculations in Weiss (2016) have quantified the interaction of ☐-NAT with water ice and have found it to be larger than for ☐-NAT which itself is more stable than ☐-NAT. There is a delicate balance of energetic terms on the way of a metastable phase to its relaxed (stable) final phase in view of the fact that the implied energy differences are small, albeit significant at the low temperatures of interest.

The aims of the present study are to investigate formation of synoptic PSC in the absence of water ice: the final paragraph of Section 1 has been revised to "Here we investigate whether MSPs and / or MFs could provide a heterogeneous nucleation pathway to NAT in synoptic PSCs where water ice is not present" in order to clarify this.

Regarding the introduction I would like to emphasize that both Grothe (2008) and Weiss (2016) dealt with heterogeneous nucleation of $HNO_3$-hydrates, reportedly in the immersion or contact freezing mode (liquid $N_2$) as well as in deposition freezing (cold metal support). The authors make it sound as if homogeneous nucleation was involved (pg. 2, line 2).
This has been corrected in the text and this section rewritten to make it clearer.

The absent proof for the existence of an (expected) effect cannot be construed as an argument in favor of its contrary! Science is deductive by nature and cannot accept negative differential diagnostics ("This must be the reason by exclusion because I cannot think of a positive fact supporting my hypothesis"). There is too much speculation owing to the absence of experimental evidence!

We have provided the first experimental evidence that meteoric materials, both smoke and fragments, have the potential to trigger nucleation of crystalline nitric acid hydrate particles in the polar stratosphere. We think this is an important conclusion and will motivate future research to generate a more quantitative and fundamentally rigorous understanding of this process in the future. The work is not speculative, since we provide clear evidence of nucleation (we observe crystallisation, which must follow on from nucleation) and justify the assumptions made when exploring the atmospheric implications of this work.

References are missing: Peter and Gross (2012); Lambert et al. (2016).

This has been corrected.

**References**

Grothe, H., Tizek, H., and Ortega, I. K.: Metastable nitric acid hydrates-possible constituents of polar stratospheric clouds?, Faraday Disc., 137, 223-234, 2008.
Höpfner, M., Luo, B. P., Massoli, P., Cairo, F., Spang, R., Snels, M., Di Donfrancesco, G., Stiller, G., von Clarmann, T., Fischer, H., and Biermann, U.: Spectroscopic evidence for NAT, STS, and ice in MIPAS

infrared limb emission measurements of polar stratospheric clouds, Atmos. Chem. Phys., 6, 1201-1219, 10.5194/acp-6-1201-2006, 2006.

Iannarelli, R., and Rossi, M. J.: The mid-IR absorption cross sections of α- and β-NAT (HNO3 · 3H2O) in the range 170 to 185 K and of metastable NAD (HNO3 · 2H2O) in the range 172 to 182 K, J. Geophys. Res.: Atmos., 120, 11,707-711,727, 10.1002/2015JD023903, 2015.

Koop, T., Luo, B. P., Tsias, A., and Peter, T.: Water activity as the determinant for homogeneous ice nucleation in aqueous solutions, Nature, 406, 611, 10.1038/35020537, 2000.

Murray, B. J., and Bertram, A. K.: Inhibition of solute crystallisation in aqueous $H^+$-$NH4^+$-$SO_4^{2-}$-$H_2O$ droplets, Phys. Chem. Chem. Phys., 10, 3287-3301, 10.1039/B802216J, 2008.

Slater, B., Michaelides, A., Salzmann, C. G., and Lohmann, U.: A blue-sky approach to understanding cloud formation, Bull. Amer. Met. Soc., 97, 1797-1802, 10.1175/bams-d-15-00131.1, 2016.

Stetzer, O., Möhler, O., Wagner, R., Benz, S., Saathoff, H., Bunz, H., and Indris, O.: Homogeneous nucleation rates of nitric acid dihydrate (NAD) at simulated stratospheric conditions; Part I: Experimental results, Atmos. Chem. Phys., 6, 3023-3033, 10.5194/acp-6-3023-2006, 2006.

Weiss, F., Kubel, F., Gálvez, Ó., Hoelzel, M., Parker, S. F., Baloh, P., Iannarelli, R., Rossi, M. J., and Grothe, H.: Metastable nitric acid trihydrate in ice clouds, Angew. Chem. Int. Ed., 55, 3276-3280, 10.1002/anie.201510841, 2016.